# Segment Policy Optimization: Effective Segment-Level Credit Assignment in RL for Large Language Models

**Yiran Guo**[†], **Lijie Xu**[†*], **Jie Liu**[†*], **Dan Ye**[†], **Shuang Qiu**[‡]

[†]Institute of Software, Chinese Academy of Sciences
[†]University of Chinese Academy of Sciences
[‡]City University of Hong Kong
{guoyiran23, xulijie, ljie, yedan}@otcaix.iscas.ac.cn
shuanqiu@cityu.edu.hk

## Abstract

Enhancing the reasoning capabilities of large language models effectively using reinforcement learning (RL) remains a crucial challenge. Existing approaches primarily adopt two contrasting advantage estimation granularities: token-level methods (e.g., PPO) aim to provide fine-grained advantage signals but suffer from inaccurate estimation due to difficulties in training an accurate critic model. On the other extreme, trajectory-level methods (e.g., GRPO) solely rely on a coarse-grained advantage signal from the final reward, leading to imprecise credit assignment. To address these limitations, we propose *Segment Policy Optimization (SPO)*, a novel RL framework that leverages segment-level advantage estimation at an intermediate granularity, achieving a better balance by offering more precise credit assignment than trajectory-level methods and requiring fewer estimation points than token-level methods, enabling accurate advantage estimation based on Monte Carlo (MC) without a critic model. SPO features three components with novel strategies: **(1)** flexible segment partition; **(2)** accurate segment advantage estimation; and **(3)** policy optimization using segment advantages, including a novel probability-mask strategy. We further instantiate SPO for two specific scenarios: **(1)** *SPO-chain* for short chain-of-thought (CoT), featuring novel cutpoint-based partition and chain-based advantage estimation, achieving 6-12 percentage point improvements in accuracy over PPO and GRPO on GSM8K. **(2)** *SPO-tree* for long CoT, featuring novel tree-based advantage estimation, which significantly reduces the cost of MC estimation, achieving 7-11 percentage point improvements over GRPO on MATH500 under 2K and 4K context evaluation. We make our code publicly available at https://github.com/AIFrameResearch/SPO.

## 1 Introduction

Reinforcement learning (RL) has become the cornerstone for training state-of-the-art reasoning large language models (LLMs), such as OpenAI o1 [11], DeepSeek R1 [7], Kimi K1.5 [31], and Qwen3 [34]. These models demonstrate RL's unique ability to cultivate advanced reasoning capabilities, particularly in complex, STEM-related tasks. However, achieving both effectiveness and efficiency in RL training hinges on addressing a fundamental challenge: *credit assignment, i.e., accurately attributing success or failure to individual actions within a sequence* [30]. In the context of LLMs, where actions correspond to generated tokens, this challenge is even greater due to sparse and delayed rewards that are typically only available at the end of the response. Advantage estimation is the common approach for credit assignment in RL, and existing methods differ in the granularity of advantage estimation, typically operating at two extremes, each with its own limitations.

---

[*]    Corresponding Authors. Lijie Xu, Jie Liu, and Dan Ye are also affiliated with Key Laboratory of System Software (Chinese Academy of Sciences) and University of Chinese Academy of Sciences, Nanjing.

39th Conference on Neural Information Processing Systems (NeurIPS 2025).

Fine-grained token-level methods like Proximal Policy Optimization (PPO) [26] use a critic model to estimate advantages for each token. However, accurately predicting state values poses a particular challenge in LLM training, due to the significant variability among states conditioned on different prompts and the limited per-prompt data to effectively train the critic model. Empirical findings by [12] provide extensive evidence that this difficulty causes the critic model to produce unreliable value predictions, leading to suboptimal credit assignment in practice. Additionally, PPO employs either a separate critic model or an additional critic head to predict the value function, leading to extra memory and computation overhead. At the other extreme, coarse-grained trajectory-level methods such as Group Relative Policy Optimization (GRPO) [28] bypass the critic model and compute a single advantage for the entire generated sequence based solely on the final outcome. While this approach is computationally efficient and unbiased, it leads to imprecise credit assignment over long sequences [12]. Applying a single advantage signal to a large number of tokens makes it challenging for the model to identify which specific tokens contribute positively or negatively, resulting in the model failing to reward partial progress or learning redundant solution paths [22]. Moreover, our experimental results, consistent with a concurrent work [35], find that GRPO can rapidly overfit on a fixed training set, with the number of unique responses decreasing and the performance on the validation set saturating early (see Figure 8).

To overcome the limitations of both token-level and trajectory-level methods, we propose *S̲egment P̲olicy O̲ptimization (SPO)*, a novel RL framework focusing on mid-grained, segment-level advantage estimation. Instead of assigning credit for each token or only at the end of a trajectory, SPO partitions the generated sequence into contiguous segments and estimates advantages at this intermediate granularity. This segment-level estimation offers several key benefits: **(1) Improved credit assignment:** Segment-level feedback provides more localized information than trajectory-level methods, allowing credit assignment to shorter segments. This finer granularity enables the model to reward partial progress even for ultimately unsuccessful responses and penalize redundancy or unnecessary portions within successful responses. **(2) More accurate advantage estimation:** Compared to token-level advantages, segment-level advantages involve fewer estimation points. This enables SPO to leverage effective Monte Carlo (MC) sampling, yielding accurate and unbiased advantage estimation directly from the policy, thus eliminating the need for an additional, unstable critic model. **(3) Flexibility and adaptability:** Our segment partition method can be arbitrarily defined without requiring semantic completeness, offering flexible adjustment of granularity from token-level to trajectory-level, also making it adaptable to a wide range of tasks.

Our SPO framework contains three key components: (**1**) **Flexible segment partition**, (**2**) **Segment advantage estimation via MC**, and (**3**) **Policy optimization using segment advantages**. This modular design allows for various strategies to be implemented within each component, making the framework highly adaptable to different tasks and scenarios. We further instantiate SPO with two specialized instances tailored for different reasoning scenarios. For short chain-of-thought (CoT), we introduce **SPO-chain**, which employs a cutpoint-based segment partition strategy and chain-based segment advantage estimation. For long CoT, we introduce **SPO-tree**, featuring a novel tree-based segment advantage estimation strategy specifically designed to significantly improve MC sampling efficiency. Additionally, we propose a novel probability-mask optimization strategy that selectively compute the loss for key tokens instead of all tokens within a segment, which can be applied to either SPO-chain or SPO-tree to further enhance credit assignment. Our experimental evaluations demonstrate the effectiveness of the SPO framework and its specialized instantiation. For short CoT, SPO-chain achieves 6–12 percentage point accuracy improvements over PPO and GRPO on GSM8K. For long CoT, SPO-tree achieves 7–11 percentage point accuracy improvements over GRPO on MATH500 under 2K and 4K context evaluation. Our major contributions are summarized as follows:

1. We propose Segment Policy Optimization (SPO), a novel segment-level RL training framework for LLMs. SPO introduces mid-grained advantage estimation to overcome the limitations of token-level and trajectory-level methods, and features a modular architecture.

2. We introduce several novel techniques integrated within the SPO framework, including cutpoint-based segment partition, tree-based segment advantage estimation, and a policy optimization strategy utilizing probability masks.

3. Building on the proposed SPO framework, we introduce two specialized instantiations: SPO-chain and SPO-tree, for short and long CoT scenarios, respectively, and demonstrate their effectiveness through extensive experiments on mathematical reasoning benchmarks, GSM8K and MATH.

## 2 Background

**Language Generation as an MDP.** Language generation tasks can be modeled as a Markov Decision Process (MDP) defined by a tuple $(\mathcal{S}, \mathcal{A}, \mathbb{P}, \mathcal{R})$, where $\mathcal{S}$ is the state space, $\mathcal{A}$ is the action space, $\mathbb{P}$ represents transition dynamics, and $R$ is the reward function. Specifically, at each time step $t$, the state $s_t \in \mathcal{S}$ consists of the prompt tokens $\boldsymbol{x}$ along with previously generated tokens $\boldsymbol{y}_{<t} = [y_1, \ldots, y_{t-1}]$, that is, $s_t = [\boldsymbol{x}, \boldsymbol{y}_{<t}]$. The action $a_t \in \mathcal{A}$ corresponds to selecting the next token $y_t$. The decision-making process follows a policy $\pi_\theta(a_t|s_t)$, parameterized by $\theta$, which defines the probability of the next token conditioned on the current state. The transition dynamics $\mathbb{P}$ are deterministic: given state $s_t$ and action (token) $a_t$, the next state $s_{t+1}$ is obtained by concatenating the selected token to the current state: $s_{t+1} = \mathbb{P}(s_t, a_t) = [s_t, a_t]$. The reward function assigns 0 intermediate rewards, providing only a sparse binary reward $\mathcal{R}(\boldsymbol{x}, \boldsymbol{y}) = 1$ or 0 at episode termination, where $\boldsymbol{y} = [y_1, y_2, \ldots, y_T]$ is the full generated response, indicating whether the generated response matches the ground truth or not. The value function $V(s)$ represents the expected cumulative reward from state $s$. The state-action value function $Q(s, a)$ corresponds to the expected cumulative reward from choosing action $a$ at state $s$. The advantage function $A(s, a)$, defined as $Q(s, a) - V(s)$, measures the improvement in expected reward realized by taking an action $a$ at state $s$. Under deterministic transition dynamics and sparse binary rewards, the advantage function simplifies to $A(s, a) = V(s') - V(s)$, where $s' = \mathbb{P}(s, a)$ is the next state reached deterministically from state $s$ after taking action $a$.

**RL for LLMs.** PPO [26, 21] and GRPO [28] are two widely adopted reinforcement learning algorithms for optimizing large language models. PPO introduces a clipped surrogate objective to stabilize training by constraining policy updates near the previous policy:

$$\mathcal{J}_{\text{PPO}}(\theta) = \mathbb{E}_{\boldsymbol{x} \sim \mathcal{D}, \boldsymbol{y} \sim \pi_{\theta_{\text{old}}}(\cdot|\boldsymbol{x})} \left[ \frac{1}{|\boldsymbol{y}|} \sum_{t=1}^{|\boldsymbol{y}|} \min\left( r_t(\theta)\hat{A}_t, \text{clip}(r_t(\theta), 1 - \epsilon, 1 + \epsilon)\hat{A}_t \right) \right], \quad (1)$$

where $r_t(\theta)$ is defined as $\frac{\pi_\theta(a_t|s_t)}{\pi_{\theta_{\text{old}}}(a_t|s_t)}$, and $\epsilon$ is a small hyperparameter that limits excessively large updates. PPO estimates token-level advantages $\hat{A}_t$ via GAE [25], requiring a critic model to predict token-level values. Compared to PPO, GRPO eliminates the need for the critic model and instead estimates the trajectory-level advantages by normalizing each response's reward within the sampled group. These trajectory-level advantages are then assigned uniformly to all tokens in the corresponding trajectory to obtain $\hat{A}_t$. Similar to PPO, GRPO adopts a clipped objective, together with a directly imposed KL penalty term. Recent work [12] highlights that accurately training the critic in PPO is challenging, and proposes VinePPO, a critic-free alternative that partitions the reasoning trajectory into discrete steps using heuristic rules (e.g., line breaks) and estimates step-level advantages through Monte Carlo (MC).

Compared to VinePPO, our proposed segment-level framework, SPO, adopts a more general concept of segmentation, allowing arbitrary partitions without enforcing semantic coherence. This enables us to freely adjust the granularity anywhere between token-level and trajectory-level, and also facilitates adaptation to broader and less structured tasks such as code generation. Moreover, we introduce a novel tree-based sampling method, where segment advantages are estimated concurrently with trajectory generation. This obviates the need for VinePPO's costly resampling to estimate advantages, thereby substantially improving efficiency and making our approach effective in long CoT scenarios. Notably, VinePPO can be viewed as a special case within our more general SPO framework. More related work is provided in Appendix A.

## 3 Segment Policy Optimization

In this section, we introduce our segment policy optimization (SPO) framework. Effective credit assignment is crucial for training LLMs in reasoning tasks. Trajectory-level methods such as GRPO rely solely on sparse final rewards, making credit assignment challenging. Token-level methods like PPO heavily depend on the critic model, whose value estimation is often inaccurate. The SPO framework aims to balance these extremes by operating at the segment granularity, enabling richer feedback signals compared to GRPO and allowing more accurate Monte Carlo estimates of segment-level advantages, thereby bypassing the need for a critic model.

We develop the SPO framework guided by the following three challenging problems: **(1)** *How to partition the generated sequence into multiple segments?* **(2)** *How to accurately and efficiently estimate the advantage for each segment?* **(3)** *How to update the policy by using the segment-level*

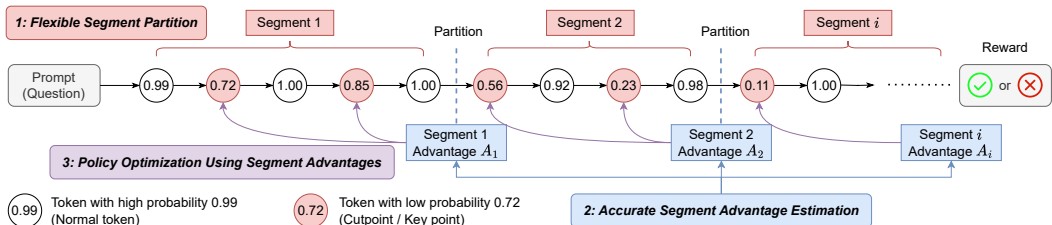

Figure 1: Overview of SPO framework. Our framework consists of three components: segment partition, segment advantage estimation, and policy optimization, each of which can be implemented in different ways. This figure illustrates the cutpoint-based partition strategy used in SPO-chain, where partitioning occurs after a predetermined number of cutpoints. It also illustrates our probability-mask policy optimization method, which assigns the corresponding segment advantages specifically to the cutpoints instead of all tokens within a segment.

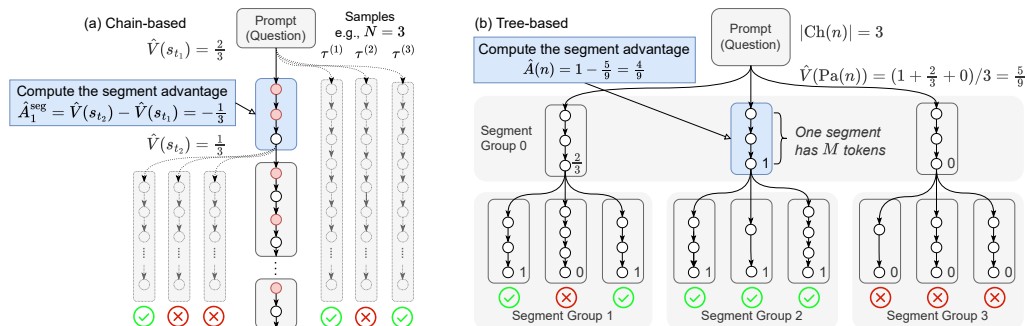

Figure 2: **(a)** Chain-based advantage estimation method. For each segment, we independently sample $N$ trajectories to estimate its value $V$. The advantage for segment $k$ is estimated as $\hat{V}(s_{t_{k+1}}) - \hat{V}(s_{t_k})$. **(b)** Tree-based advantage estimation method. Trajectories are organized in a tree structure, where nodes sharing the same parent form a group with identical prompts and token counts (except for leaf nodes, whose token lengths may vary). This hierarchical organization facilitates the calculation of advantages within each group.

*advantage?* The proposed SPO framework, as illustrated in Figure 1, adopts a modular architecture, where each of its components can be implemented using various strategies, allowing the framework to be tailored to diverse tasks and scenarios. In what follows, we introduce three key components of SPO to answer the above questions and further instantiate SPO with two instances, tailored for short and long CoT scenarios.

**(1) Flexible Segment Partition.** A segment is defined as a contiguous sequence of generated tokens, denoted as $\text{seg}_k = [y_{t_k}, y_{t_k+1}, \ldots, y_{t_{k+1}-1}]$, where $t_k$ is the starting token index of the $k$-th segment. Formally, given a full generated trajectory $\boldsymbol{y} = [y_1, y_2, \ldots, y_T]$, the partition can be expressed as $\boldsymbol{y} = [y_1, y_2, \ldots, y_T] = [\text{seg}_1, \text{seg}_2, \cdots, \text{seg}_K]$. The SPO framework supports arbitrary partition strategies, allowing flexible definition of segment boundaries, without requiring semantic completeness. This flexibility enables us to choose partition granularity between token-level and trajectory-level, allowing a trade-off between computational cost and credit assignment. In this work, we consider two main partition strategies, designed for different scenarios: **(a) Fixed Token Count Partition:** A straightforward strategy that divides the sequence into segments of a predetermined fixed number of tokens. **(b) Adaptive Cutpoint-based Partition:** An advanced strategy (see Section 4) that defines segments by accumulating a fixed number of low probability tokens (i.e., tokens whose probabilities $\pi_\theta(y_t|s_t)$ are less than a threshold $\rho$). This strategy places segment boundaries at positions where $V$ values are more likely to change, avoiding the issue in the fixed token count partition strategy where $V$ values may remain unchanged between segment boundaries when each segment is short.

**(2) Segment Advantage Estimation via Monte Carlo.** After obtaining each segment $\text{seg}_k = [y_{t_k}, y_{t_k+1}, \ldots, y_{t_{k+1}-1}]$, we define the segment advantage in the following way:

$$A_k^{\text{seg}} := V(s_{t_{k+1}}) - V(s_{t_k}) = V([\boldsymbol{x}, \text{seg}_1, \ldots, \text{seg}_k]) - V([\boldsymbol{x}, \text{seg}_1, \ldots, \text{seg}_{k-1}]), \qquad (2)$$

which measures the incremental improvement in expected reward resulting from generating the tokens in $\text{seg}_k$. The core challenge lies in accurately estimating the value function $V(s)$. A common approach is to train a critic model. However, in the LLM scenario, critic-based methods struggle to produce accurate state-value estimates due to large variations across prompts and limited trajectory data per prompt, as empirically illustrated by [12]. Given these considerations, we propose to bypass the critic entirely and instead adopt Monte Carlo (MC) estimation to compute unbiased estimates of segment values directly from sampled trajectories. While high variance is often a concern with MC estimation, in the LLM setting, rewards are typically sparse and binary, significantly mitigating the variance issue. Specifically, we can view $V$ as a Bernoulli variable, and the MC variance is at most 0.25. We find that the accuracy obtained with a small number of samples ($N = 4$ or $N = 9$) is sufficient for effective policy optimization. Our SPO framework supports various MC sampling strategies: **(a) Chain-based Advantage Estimation:** Independently roll out $N$ trajectories from state $s$, and estimate $V(s)$ as the average reward of these trajectories, further yielding the estimation of the advantage $A_k^{\text{seg}}$ following Equation (2). **(b) Tree-based Advantage Estimation:** An advanced strategy of advantage estimation for improved sample efficiency (see details in Section 5). Unlike the chain-based advantage estimation, which discards MC samples after estimating the $V$ values, this strategy organizes MC samples into a tree structure and computes the $V$ values via bottom-up aggregation of rewards along the tree. Nodes sharing the same parent have identical prompts and token counts, forming segment groups, enabling advantage computations within each group. This tree structure allows the reuse of MC samples for policy optimization, significantly enhancing sample efficiency in the long CoT scenario.

**(3) Policy Optimization Using Segment Advantages.** Given a set of generated segments $\{\text{seg}_k\}$ and their corresponding advantages $\{A_k^{\text{seg}}\}$, we now focus on effectively leveraging training samples to update the policy $\pi_\theta$. Specifically, we can adapt various policy optimization approaches into our SPO framework: **(a) Policy Gradient:** We can assign $A_k^{\text{seg}}$ to all tokens contained within the $k$-th segment $\text{seg}_k$ and then optimize using the PPO loss; **(b) Policy Gradient with Token-Probability Masks:** An improved version of policy gradient (see details in Section 4), incorporating probability masks. Instead of uniformly assigning $A_k^{\text{seg}}$ to all tokens within the segment, this strategy exclusively assigns $A_k^{\text{seg}}$ only to low-probability tokens, based on the intuition that these tokens primarily contribute to the segment's advantage. This refined approach further enhances the credit assignment to critical tokens; **(c) Other Strategies:** Our framework can also be adapted to other policy optimization methods such as policy iteration-based approach (see discussions in Appendix B) and potentially GFlowNet-based optimization [2].

## 4 SPO-Chain for Short CoT

For short Chain-of-Thought (CoT) scenarios, where the computational overhead of MC sampling is low and segments typically involve a small number of tokens, we designed a tailored instance of SPO by taking into account these characteristics, which we refer to as **SPO-chain**. The core features of SPO-chain lie in its cutpoint-based segment partition, chain-based segment advantage estimation, and policy optimization via policy gradient with probability masks.

**(1) Adaptive Cutpoint-based Segment Partition.** The fixed token count partition strategy faces a critical issue in short CoT scenarios that the number of tokens in each segment is not very large. If the token probabilities $\pi_\theta(y_t|s_t)$ within a segment are high (close to 1), the $V$ values estimated at the beginning and end of the segment will not differ significantly. This can lead to unnecessary partitioning, ultimately resulting in a waste of sampling budget. To address this, we propose an adaptive cutpoint-based partition strategy. Specifically, we first identify the *cutpoints*, defined as positions where token probabilities drop below a pre-defined threshold $\rho$, so that we have the set of all cutpoints defined as $\mathcal{U}_\theta = \{t < T \mid \pi_\theta(y_t|s_t) < \rho\}$. These cutpoints represent positions where the model's reasoning trajectory could diverge, thus potentially inducing changes in $V$ values. For better credit assignment, we prefer each segment to contain fewer cutpoints. Then, given a fixed segment count $K$, we find a partition that reflects this principle by solving the following optimization problem: $\min_{\{t_k\}_{k=1}^K} \sum_{k=1}^K |\mathcal{U}_\theta \cap [t_k, t_{k+1})|^2$. It can be shown that the optimal solution evenly distributes cutpoints across segments (assuming $|\mathcal{U}_\theta|$ is divisible by $K$ for simplicity), that is,

$$|\mathcal{U}_\theta \cap [t_k, t_{k+1})| = \frac{|\mathcal{U}_\theta|}{K}, \qquad \forall k \leq K$$

which corresponds to partitioning the trajectory such that each segment contains the same number of cutpoints, as shown in Figure 1. A practical example is provided in Appendix I. Our experiments

further show that this partition strategy would lead to a superior performance (see our comparison of different segment partition strategies in Section 6).

**(2) Chain-based Segment Advantage Estimation.** In the short CoT scenario, the computational overhead of MC estimation is generally manageable. Thus, we adopt a simple, chain-based MC sampling approach as shown in Figure 2(a). Specifically, at each segment boundary state $s_{t_k} = [\boldsymbol{x}, \boldsymbol{y}_{<t_k}]$, we independently sample $N$ trajectories from the policy $\pi_\theta$, i.e. $\boldsymbol{\tau}_{t_k}^{(j)} \sim \pi_\theta(\cdot|s_{t_k}), j = 1, \ldots, N$ and then estimate the value of such a state by averaging the returns from these sampled trajectories, i.e., $\hat{V}(s_{t_k}) = \frac{1}{N} \sum_{j=1}^{N} \mathcal{R}(\boldsymbol{x}, [\boldsymbol{y}_{<t_k}, \boldsymbol{\tau}_{t_k}^{(j)}])$. The estimated advantage $\hat{A}_k^{\text{seg}}$ of each segment $k$ is computed by taking the difference between the state values at consecutive segment boundaries $\hat{A}_k^{\text{seg}} = \hat{V}(s_{t_{k+1}}) - \hat{V}(s_{t_k})$ following Equation (2).

**(3) Policy Gradient with Token-Probability Masks.** Policy gradient methods have become mainstream for training LLMs using RL. Accordingly, we adopt a policy gradient-based approach to optimize the policy. For example, once we have computed the segment advantage $A_k^{\text{seg}}$, we can assign it to all tokens within the segment and obtain the token-level advantage $A_t$, then adopt the PPO loss in Equation (1) to optimize the policy. However, since the changes in $V$ values between segments are primarily caused by tokens at cutpoints, we assign the segment advantage $A_k^{\text{seg}}$ only to these critical tokens. Formally, our policy is trained by minimizing the following loss:

$$
\mathcal{J}_{\text{SPO}}^{\text{chain}}(\theta) = \mathbb{E}_{\boldsymbol{x} \sim \mathcal{D}, [\text{seg}_1, \ldots, \text{seg}_K] \sim \pi_{\theta_{old}}}
$$
$$
\left\{ \frac{1}{Z} \sum_{k=1}^{K} \sum_{t=t_k}^{t_{k+1}-1} \left[ M_t \cdot \min\left( r_t(\theta)\hat{A}_k^{\text{seg}}, \text{clip}\big(r_t(\theta), 1-\epsilon, 1+\epsilon\big)\hat{A}_k^{\text{seg}} \right) - \beta D_{\text{KL}}(\pi_\theta \| \pi_{\text{ref}}) \right] \right\}, \tag{3}
$$

where $M_t$ is the token probability mask whose value is 1 if $\pi_{\theta_{old}}(a_t|s_t) < \rho$ and 0 otherwise, i.e., $M_t := \mathbb{I}\big(\pi_{\theta_{old}}(a_t|s_t) < \rho\big)$. In addition, $Z$ is a normalization term that equals the total number of tokens where the mask $M_t$ is 1: $Z = \sum_{t=1}^{T} M_t$. This approach further enhances credit assignment within each segment by concentrating the advantage signals on fewer critical tokens. Our experiments demonstrate that this method improves the accuracy (refer to our ablation study on the probability-mask optimization strategy in Section 6).

## 5 SPO-Tree for Long CoT

For long Chain-of-Thought (CoT) scenarios, where the sampling cost of chain-based MC estimation in SPO-chain becomes prohibitively high, we propose a novel tree-based segment advantage estimation method. This approach drastically reduces sampling overhead, making it feasible to apply our framework effectively in long CoT settings. We refer to this instantiation of our SPO framework as **SPO-tree**. In particular, SPO-tree features the fixed token count partition, tree-based segment advantage estimation, and policy gradient with probability masks.

**(1) Fixed Token Count Segment Partition.** In long CoT scenarios, each segment typically contains a large number of tokens. As a result, it is unlikely that all token transitions within an entire segment will have probabilities close to 1. Thus, we employ a partition strategy where segment boundaries are made at fixed token intervals, which allows us to generate segments with equal token count, supporting our tree-based segment advantage estimation method proposed in the following subsection.

**(2) Tree-based Segment Advantage Estimation.** The core drawback of the chain-based segment advantage estimation strategy studied in Section 4 is that it discards samples used for estimating the values (e.g., $\boldsymbol{\tau}^{(1)}, \boldsymbol{\tau}^{(2)}, \boldsymbol{\tau}^{(3)}$ in Figure 2(a)), leading to substantial waste of samples in scenarios involving long reasoning trajectories. To address this issue, we propose a tree-based segment advantage estimation strategy, which reuses the samples employed for value estimation in policy optimization, significantly improving the sample efficiency.

As illustrated in Figure 2(b), we model the trajectory sampling process as a tree structure. We define $\text{hist}(n)$ as the full history sequence corresponding to node $n$, including the initial prompt $\boldsymbol{x}$ and all tokens generated along the path from the root node to node $n$. Each node $n$ represents a segment $\text{seg}(n)$, which is generated by extending the sequence of its parent node, $\text{Pa}(n)$, with $M$ newly sampled tokens, i.e.,

$$
\text{seg}(n) = \big[y_1^{(n)}, \ldots, y_M^{(n)}\big], \quad y_t^{(n)} \sim \pi\big(\cdot \mid \big[\text{hist}(\text{Pa}(n)), \boldsymbol{y}_{<t}^{(n)}\big]\big).
$$

Each node $n$ generates a set of child nodes, denoted as $\mathrm{Ch}(n)$, and has a set of siblings, denoted as $\mathrm{Sib}(n)$. Nodes among siblings share the same prompts and sequence lengths (except for leaf nodes), ensuring fair comparison under the same token budget. The value of the state represented by node $n$ is denoted as $V(n)$, which can be estimated recursively from leaves to root as follows:

$$\hat{V}(n) = \begin{cases} \mathcal{R}(\boldsymbol{x}, \mathrm{hist}(n)), & \text{if } n \text{ is a leaf node} \\ \frac{1}{|\mathrm{Ch}(n)|} \sum_{n' \in \mathrm{Ch}(n)} \hat{V}(n'), & \text{otherwise} \end{cases}.$$

We denote the advantage of the segment represented by node $n$ as $A(n)$. The estimated advantage $\hat{A}(n)$ for node $n$, relative to its siblings, is computed either in unnormalized or normalized form as:

$$\hat{A}(n) = \hat{V}(n) - \texttt{mean}_{n' \in \mathrm{Sib}(n)} \hat{V}(n') \quad \text{or} \quad \frac{\hat{V}(n) - \texttt{mean}_{n' \in \mathrm{Sib}(n)} \hat{V}(n')}{\texttt{std}_{n' \in \mathrm{Sib}(n)} \hat{V}(n')}.$$

More details about the tree construction are presented in Appendix E. In contrast to the chain-based segment advantage estimation strategy, each node (segment) in the tree can be used as a training example, substantially improving sample efficiency. In addition, due to extensive node-sharing among trajectories, the actual number of tokens we need to sample is significantly less than the sum of tokens across all trajectories, greatly reducing sampling overhead. Within each training iteration, sampling a large number of trajectories from a single question can lead to the model overfitting to specific samples. To address this, we introduce a replay buffer mechanism that distributes sampled trajectories across multiple iterations in Appendix D.

The width of the tree determines the number of sibling nodes in each group, with a larger width providing more nodes for comparison and thus more nuanced advantage estimates. The depth of the tree controls the granularity of partitions; deeper trees yield finer-grained signals. Furthermore, the hierarchical tree structure naturally aligns with the varying uncertainty along the reasoning trajectory. Earlier segments (closer to the root), which have higher uncertainty, aggregate more samples for estimating their $V$ values, while later segments (closer to the leaves), which are more certain, aggregate fewer samples.

**(3) Policy Gradient with Token-Probability Masks.** To focus model training on segments with discriminative signals, we extract segments with non-zero advantages from the tree for training. We also note that such filtering of non-zero advantage samples aligns with practices in several concurrent works [35, 15]. Thus, we propose to minimize $\mathcal{J}_{\mathrm{SPO}}^{\mathrm{tree}}$ below for policy optimization:

$$\mathcal{J}_{\mathrm{SPO}}^{\mathrm{tree}}(\theta) = \mathbb{E}_{\boldsymbol{x} \sim \mathcal{D}, \, \mathrm{tree} \sim \pi_{\theta_{\mathrm{old}}}} \left\{ \frac{1}{Z} \sum_{n \in \mathrm{tree}: \hat{A}(n) \neq 0} \sum_{t=1}^{|\mathrm{seg}(n)|} \right.$$
$$\left. \left[ M_{n,t} \cdot \min\left( r_{n,t}(\theta)\hat{A}(n), \, \mathrm{clip}(r_{n,t}(\theta), 1-\epsilon, 1+\epsilon)\hat{A}(n) \right) - \beta D_{\mathrm{KL}}(\pi_\theta \| \pi_{\mathrm{ref}}) \right] \right\}, \tag{4}$$

where $M_{n,t}$ is the token-probability mask whose value is 1 if $\pi_\theta(y_t^{(n)} \mid [\mathrm{hist}(\mathrm{Pa}(n)), \boldsymbol{y}_{<t}^{(n)}]) < \rho$ and 0 otherwise, i.e., $M_{n,t} := \mathbb{I}\left( \pi_\theta(y_t^{(n)} \mid [\mathrm{hist}(\mathrm{Pa}(n)), \boldsymbol{y}_{<t}^{(n)}]) < \rho \right)$. The ratio term $r_{n,t}(\theta)$ is then defined as $\frac{\pi_\theta(y_t^{(n)} \mid [\mathrm{hist}(\mathrm{Pa}(n)), \boldsymbol{y}_{<t}^{(n)}])}{\pi_{\theta_{\mathrm{old}}}(y_t^{(n)} \mid [\mathrm{hist}(\mathrm{Pa}(n)), \boldsymbol{y}_{<t}^{(n)}])}$, and the normalization term $Z$ is given by $Z = \sum_{n \in \mathrm{Tree}: \hat{A}(n) \neq 0} \sum_{t=1}^{|\mathrm{seg}(n)|} M_{n,t}$.

## 6 Experiments on SPO-chain

**Experimental Setups.** We evaluate SPO-chain with the RhoMath 1.1B model [14] on the GSM8K dataset [5] using Pass@1 (accuracy) as our primary evaluation metric. Following [12], we first finetune the base model on the GSM8K training set, and then use the obtained SFT model for subsequent RL training. We compare against baseline methods including RestEM, DPO, PPO, GRPO, RLOO, and VinePPO. The experimental hyperparameters largely follow [12], with detailed settings provided in Appendix H. For SPO-chain, we set the number of MC samples to $N = 9$ and partitioned the model output at **intervals of every 5 cutpoints**, which is referred to as **SPO-chain (int5)**. We also evaluate our SPO-tree method in the short CoT scenario. We adopt a 6-6-6 tree structure, meaning

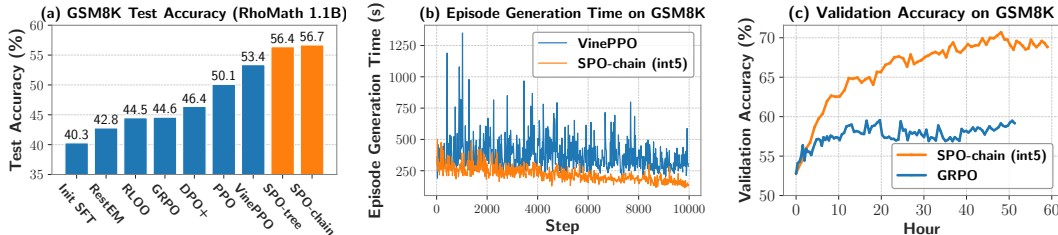

Figure 3: **(a)** Test accuracy comparison of different methods on GSM8K. Baseline results are from [12]. **(b)** Episode generation time comparison between SPO-chain (int5) and VinePPO during training. **(c)** Validation accuracy of SPO-chain (int5) and GRPO during training.

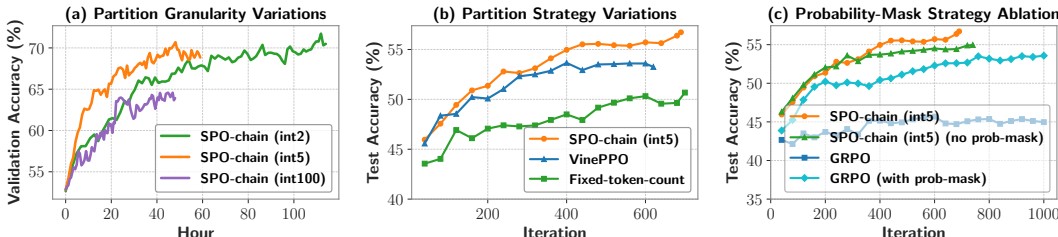

Figure 4: **(a)** Variations of segment partition granularity (different cutpoint intervals). **(b)** Variations of segment partition strategies. **(c)** Ablation on probability-mask policy optimization strategy.

the tree consists of three layers and each internal node expands into six child nodes, which is referred to as **SPO-tree (6-6-6)**. For the first two layers, we segment the model output every 30 tokens, while for the final layer, we let the model complete the entire reasoning trajectory.

**Comparison with Baseline Methods.** As shown in Figure 3(a), our method, SPO-chain (int5), achieves the highest accuracy on the GSM8K test set, outperforming PPO and GRPO by 6-12 percentage points. Furthermore, compared to VinePPO, our method not only delivers higher accuracy but also requires less time for episode generation, as demonstrated in Figure 3(b). This is due to our SPO-chain (int5) method uses fewer advantage estimation points per trajectory compared to VinePPO. Additionally, our SPO-tree (6-6-6) method achieves the second-highest accuracy on the GSM8K test set, validating its effectiveness and making it an efficient alternative. We also compared our method with GRPO in terms of validation performance under the same wall-clock time[2] (Figure 3(c)). The results show that our algorithm significantly outperforms GRPO and ultimately converges to a much better solution. We further conduct experiments with additional models, including DeepSeekMath 7B [28], Qwen base and instruct models [34], as well as on the non-mathematical Knights-and-Knaves dataset [33]. The detailed results of these experiments are presented in Appendix F.

**Impact of Segmentation Granularity.** To investigate how the segment granularity affects model performance, we conducted experiments using various segment intervals, as shown in 4(a). When comparing validation accuracy under the same wall-clock time, interval 5 achieves the highest performance, followed by interval 2, while interval 100 performs the worst. In terms of final accuracy, interval 2 slightly surpasses interval 5, whereas interval 100 lags significantly behind both. These results indicate that a moderate interval (e.g., 5) provides a more favorable trade-off, yielding better accuracy given the same training time, while excessively fine-grained intervals (e.g., 2) bring only marginal improvements and overly coarse intervals severely harm the accuracy. This supports the design of our segment-level advantage method, which strikes an effective balance between efficiency and accuracy, achieving substantially better performance than trajectory-level advantage without the high computational cost of token-level estimation.

**Comparison of Different Segment Partition Strategies.** We compare different trajectory partition methods, including the naive fixed token count partition, the heuristic rules (e.g. line breaks) for partitioning as in VinePPO, and the cutpoint-based segment partition in SPO-chain. The results are presented in Figure 4(b). For the naive fixed token count partitioning method, we explicitly set each trajectory's segment count at 3, making it higher than the total sampling budget of the SPO-chain

---

[2]Evaluation time is also included in the reported wall-clock time. The model is evaluated every 10 iterations.

(int5). Remarkably, despite having the smallest sampling budget, SPO-chain (int5) achieves the best test accuracy, validating the effectiveness of our cutpoint-based segment partition strategy.

**Ablation Study on Probability-Mask Optimization Strategy.** We conduct an ablation study on our proposed probability-mask optimization technique. As shown in Figure 4(c), removing the probability-mask technique leads to a decreased accuracy for SPO-chain (int5), from 56.7% to 55.0%. Surprisingly, applying the probability-mask technique to GRPO results in a significant improvement, boosting its accuracy from 45.7%[3] to 53.6%. We hypothesize that employing the probability-mask technique enables more precise credit assignment to tokens that are most likely to influence the model's reasoning trajectory. Meanwhile, the losses associated with tokens whose probabilities are close to 1 are masked out and thus excluded from the overall loss function, which potentially helps reduce overfitting.

# 7 Experiments on SPO-tree

**Experimental Setups.** To evaluate our method in long CoT scenario, we utilize DeepSeek-R1-Distill-Qwen-1.5B model as our base model, which has been fine-tuned on huge amount of long CoT data, demonstrating complex reasoning strategies and typically generating long outputs. Given the strong capability of these models, we select the competition-level MATH [10] dataset for our training and evaluation. We mainly focus on comparing our proposed SPO-tree method against GRPO, as GRPO is the mainstream training approach adopted for reasoning models, particularly favored for long CoT scenarios due to its efficiency. We do not compare with VinePPO because its efficiency issue, making it impractical to apply in long CoT settings. For our SPO-tree method, the structure remains identical to the SPO-tree used in the short CoT scenario, as described in Section 6. Detailed hyper-parameters can be found in Appendix H.

**Comparison with Baseline Methods.** We initially limit the training context window size to 2K and evaluate model accuracy (Pass@1) on MATH500 every 10 iterations. For evaluation, we follow the implementation provided by [12], setting the evaluation context window also to 2K and using greedy decoding (temperature = 0). Figure 5(a) shows the training curves of SPO-tree, vanilla GRPO, and GRPO with probability mask. We observe that GRPO with probability-mask outperforms vanilla GRPO to some extent, demonstrating the effectiveness of this technique in long CoT scenarios. However, SPO-tree still achieves significantly higher accuracy under the same wall-clock time. This validates the effectiveness of SPO-tree and highlights that introducing segment advantage values is crucial for achieving better performance.

To further assess our method, we continue training from the 2K checkpoint and expand the model's context length to 4K. The corresponding model performance is presented in Table 1. We adopt the evaluation script provided by [8] to conduct evaluation on the obtained model checkpoints. As shown in the results, the SPO-tree consistently outperforms GRPO across all context sizes and achieves superior performance under 2K and 4K contexts, which are the context lengths it was trained on. We hypothesize that our segment-level advantage method significantly increases token efficiency due to more precise credit assignment, enabling the model to arrive at correct answers more straightforwardly (see Appendix J). This improvement leads to considerably better performance under limited context sizes. Interestingly, even though our model is trained under context windows up to 4K, it still achieves improved performance under 32K context evaluation, surpassing the base model. We also compared our approach against the most recent works, including DeepScaleR [18] and STILL-3 [4], both of which were trained on DeepSeek-R1-Distill-Qwen-1.5B using GRPO. Notably, DeepScaleR and STILL-3 adopted larger training datasets and extended context windows for training. Specifically, DeepScaleR progressively increased the context length from 8K to 16K and finally to 24K, whereas our training scales only from 2K to 4K. While DeepScaleR performs best at 32K, we observe the opposite trend in lower-context settings, where DeepScaleR exhibits the worst performance at 2K and 4K. This finding suggests that GRPO-based training might not optimize token efficiency, potentially leading to redundancy in the generated outputs, and thus negatively affecting the model's performance when evaluated under limited context size.

---

[3]Our GRPO implementation achieves a slightly higher test accuracy (45.7%) compared to the 44.6% reported in [12].

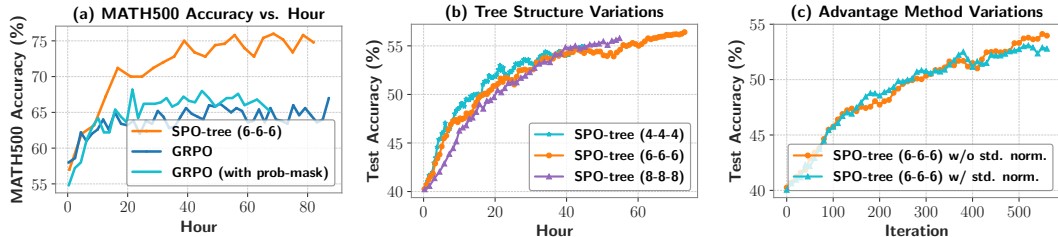

Figure 5: **(a)** Comparison of SPO-tree (6-6-6) and GRPO on MATH500 with a context size of 2K. **(b)** Variations of tree structures on GSM8K. **(c)** SPO-tree with different advantage methods on GSM8K.

| Dataset | Eval Context Size | Base | GRPO | SPO-tree | DeepScaleR[*] | STILL-3[*] |
|---------|-------------------|------|------|----------|---------------|-----------|
| MATH500 | 2K | 0.566 | 0.62 | **0.736** | 0.538 | 0.662 |
|         | 4K | 0.74 | 0.752 | **0.828** | 0.744 | 0.794 |
|         | 32K | 0.838 | 0.84 | 0.848 | **0.878** | 0.846 |
| AIME24 | 2K | 0.067 | 0.033 | **0.1** | 0 | 0.067 |
|        | 4K | 0.167 | **0.2** | **0.2** | 0.167 | 0.133 |
|        | 32K | 0.267 | **0.333** | **0.333** | **0.333** | 0.233 |

[*] **These models were trained using larger datasets and longer context windows**. Specifically, DeepScaleR uses 8K→16K→24K context and STILL-3 sets 'generate_max_len' to 29000 during training, while our GRPO and SPO-tree use a maximum context of 4K during training.

Table 1: Accuracy comparison among methods with various context sizes on MATH500 and AIME24.

**Comparison of Different Tree Structures and Advantage Computation Methods.** To investigate the impact of different tree structures, we compared the performance[4] of various tree structures (4-4-4, 6-6-6, 8-8-8) on the GSM8K test set (Figure 5(b)). When using a larger tree structure, each iteration generates a greater number of segments with non-zero advantage. Therefore, we set different values of the hyperparameter "num_episodes_per_iteration" for different tree structures: specifically, we set it to 1024 for SPO-tree (8-8-8), 512 for SPO-tree (6-6-6), and 384 for SPO-tree (4-4-4). We found that the performance differences among these tree structures were not substantial under the same wall-clock time, indicating the robustness of the tree structure. Smaller tree structures achieve higher accuracy initially, possibly because they can process more data examples within the same time frame. However, larger tree structures eventually outperform smaller ones at later training stages, as they enable more accurate value estimation and benefit from having more segments within each group, leading to more reliable and nuanced advantage estimates. We also compared the performance of different advantage calculation methods with and without standard deviation normalization. As shown in Figure 5(c), both methods perform similarly, while the variant without normalization performs slightly better.

## 8   Conclusion

In this work, we propose Segment Policy Optimization (SPO), a novel RL training framework for large language models (LLMs) that effectively addresses the limitations of existing RL approaches. SPO provides feedback at the segment level, offering finer granularity than trajectory-level methods for more precise credit assignment, while reducing the number of advantage estimation points compared to token-level approaches. This enables us to leverage Monte Carlo methods to obtain unbiased advantage estimates. Our experiments demonstrate that a small number of segment-level advantages can significantly outperform coarse trajectory-level advantages, validating the effectiveness of our framework. Due to limited computational resources, our current experiments in the long CoT scenario are limited to a maximum context size of 4K tokens. In the future, we plan to conduct additional experiments with larger context sizes. Additionally, our current experiments focus primarily on mathematical tasks. We plan to test the effectiveness of SPO in broader application scenarios, such as code generation and RLHF.

---

[4]Evaluation time is also included in the reported wall-clock time. The model is evaluated every 10 iterations.

## Acknowledgments and Disclosure of Funding

This work was supported by the Major Project of ISCAS (ISCAS-ZD-202302), the National Natural Science Foundation of China (Grant No. 42361144884), and the General Research Fund (GRF 16209124). We would like to thank MiraclePlus for providing the computational resources. The authors declare no competing interests.

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

# A Detailed Related Work

**Boosting LLM's Reasoning Capacity**. Various approaches have been explored to strengthen the reasoning abilities of LLMs. Several methods focus on using high-quality data for training. For instance, RFT [36, 37] fine-tunes the pretrained base model using correct samples generated by a supervised fine-tuned model. Similarly, RestEM [29] adopts an iterative self-training strategy, repeatedly retraining the base model on high-quality reasoning traces produced by previously obtained checkpoints. Preference-based methods like DPO [24, 23] optimize models by contrasting correct and incorrect reasoning outputs. Search-guided approaches, like Monte Carlo Tree Search (MCTS) [38, 3], help models discover improved reasoning paths during inference. Another significant direction involves RL frameworks, which typically adopt policy gradient-based methods and differ mainly in advantage estimation granularity. For example, GRPO [28] and RLOO [1] estimate trajectory-level advantages, while PPO [26, 21] estimates token-level advantages using a dedicated critic model. Recent work [12] highlights that accurately training the critic in PPO is challenging, and proposes VinePPO, a critic-free alternative that partitions the reasoning trajectory into discrete steps using heuristic rules (e.g., line breaks) and estimates step-level advantages through Monte Carlo (MC) sampling. PRIME [6] simultaneously trains a process reward model using outcome rewards during policy training, providing process rewards to better guide policy optimization, while our work does not rely on reward models. Most recently, several works have proposed improvements upon the original GRPO method, like DAPO [35] and Dr. GRPO [16], based on the trajectory-level advantage estimation.

**Fine-Grained Reward Signals**. A common approach in prior work assigns a single binary reward to the final output of LLMs by comparing it against the ground truth. However, this sparse reward provides limited guidance, resulting in the difficulty of credit assignment during training. To address this challenge, [13] initially proposed "Process Reward", which involves manually judging correctness at every intermediate reasoning step. Subsequent works, such as [32], [17], and [19], automated the process reward generation without manual labeling through rollouts. These methods focus on training an external Process Reward Model (PRM) to provide intermediate rewards for policy optimization and enable Best-of-N (BoN) selection during inference. In practice, when applying PRM for policy optimization, existing approaches typically combine step-level rewards with the final binary correctness rewards and still employ standard RL algorithms such as PPO or GRPO for training. In contrast, our approach differs fundamentally by improving the RL optimization algorithm itself, rather than relying on external reward models. We introduce an MC-based estimation method to directly compute segment-level advantages from the current policy, aiming at potentially reducing gradient estimation variance and enabling finer-grained credit assignment during optimization. This strategy helps stabilize training and leads to more effective policy updates compared to using sparse, trajectory-level rewards alone. Notably, our MC advantage estimation framework remains fully compatible with the PRM approach: intermediate process reward signals generated via PRMs could be integrated with our trajectory evaluations. Such integration may combine the strengths of both methodologies, providing further potential improvements in overall model performance. In Appendix C, we provide an analysis of integrating the process reward into our framework.

# B Policy Learning via Policy Iteration

We can alternatively adopt a policy iteration-based approach formulated in the following way for policy learning. The objective of training LLMs via RL can be written as a KL-constrained policy optimization problem:

$$J(\pi) = \mathbb{E}_\pi \left[ \sum_{h=1}^{H} \left( r(s_h, a_h) - \beta \log \frac{\pi(a_h|s_h)}{\pi_{\text{ref}}(a_h|s_h)} \right) \right].$$

By defining the modified reward $\bar{r}(s, a) = r(s, a) + \beta \log \pi_{\text{ref}}(a|s)$, we arrive at an equivalent maximum-entropy RL formulation:

$$J(\pi) = \mathbb{E}_{\pi, s_0 \sim d_0} \left[ \sum_{h=1}^{H} \left( \bar{r}(s_h, a_h) + \beta \mathcal{H}(\pi(\cdot|s_h)) \right) \right].$$

Under this formulation, the Q-function and value function satisfy the soft Bellman equation:

$$Q(s_h, a_h) = \bar{r}(s_h, a_h) + V(s_{h+1}).$$

Given the above relationships, we can optimize the Q-function via the loss:

$$L_Q(\theta) = \mathbb{E}_{(s_h,a_h,s_{h+1})\sim\mathcal{D}} \left[ (Q_\theta(s_h,a_h) - \bar{r}(s_h,a_h) - V_\phi(s_{h+1}))^2 \right].$$

Given the estimates of Q-function and value function, the optimal policy satisfies:

$$Q_\theta(s_h,a_h) = V_\phi(s_h) + \beta \log \pi_\theta(a_h|s_h).$$

Substituting this optimal policy relationship back into the Bellman equation results in the following policy optimization objective:

$$L_\pi(\theta) = \mathbb{E}_{(s_h,a_h,s_{h+1})\sim\mathcal{D}} \left[ (V_\phi(s_h) + \beta \log \pi_\theta(a_h|s_h) - \bar{r}(s_h,a_h) - V_\phi(s_{h+1}))^2 \right].$$

Since we have estimated the advantages $A(s_h,a_h) = V(s_{h+1}) - V(s_h)$, explicit estimation of value function $V$ is not necessary. Thus, the policy optimization objective further simplifies to:

$$L_\pi(\theta) = \mathbb{E}_{(s_h,a_h)\sim\mathcal{D}} \left[ \left( \beta \log \frac{\pi_\theta(a_h|s_h)}{\pi_{\text{ref}}(a_h|s_h)} - A(s_h,a_h) \right)^2 \right].$$

Each iteration of this procedure corresponds to a policy improvement step, and after updating the policy, newly sampled trajectories can be employed to update the Q-function estimation.

We conduct preliminary experiments using this objective function, where we adopt our SPO-tree (6-6-6) method and introduce an importance weight $\frac{\pi_\theta}{\pi_{old}}$ to alleviate the off-policy influence. The results, shown in Figure 6, demonstrate that our approach enables steady policy improvement. However, it does not reach the performance achieved by policy gradient with probability masks described in Section 5), which achieves a validation accuracy slightly above 0.7.

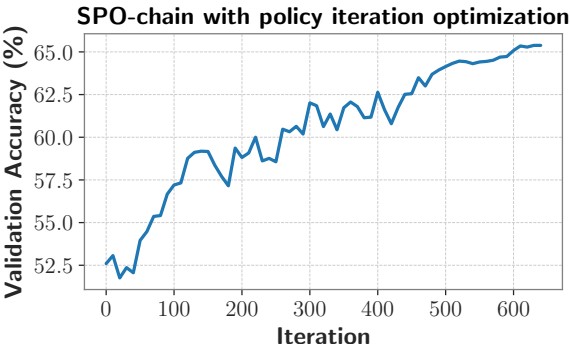

Figure 6: Validation accuracy on GSM8K using policy iteration optimizing method.

## C    Incorporate Process Reward

Our framework is compatible with the process reward. Inspired by the work in [27], we adopt the current policy's BoN as the prover policy $\mu$ and utilize its advantage as the process reward. Specifically, the gradient is computed as follows:

$$\nabla_\pi \ell(\pi) = \sum_{h=1}^{H} \nabla_\pi \log \pi(a_h \mid s_h) \cdot (Q^\pi(s_h,a_h) + \alpha \cdot A^\mu(s_h,a_h))$$

Unlike the aforementioned work, our method does not require training a separate reward model (PAV). Given that the adopted prover policy $\mu$ is the current policy's BoN, we can compute its value function $V^\mu(s_h)$ from the value function of the current policy $V^\pi(s_h)$:

$$V^\mu(s_h) = 1 - (1 - V^\pi(s_h))^N,$$

where $N$ is the number of MC samples. Then, the prover policy's advantage $A^\mu(s_h,a_h)$ can be easily computed as follows:

$$A^\mu(s_h,a_h) = V^\mu(s_{h+1}) - V^\mu(s_h).$$

Thus, we can directly integrate process rewards into our framework using MC samples generated from the current policy, simplifying the overall algorithm and effectively circumventing potential issues that arise from introducing a separate reward model, such as fitting errors and reward hacking.

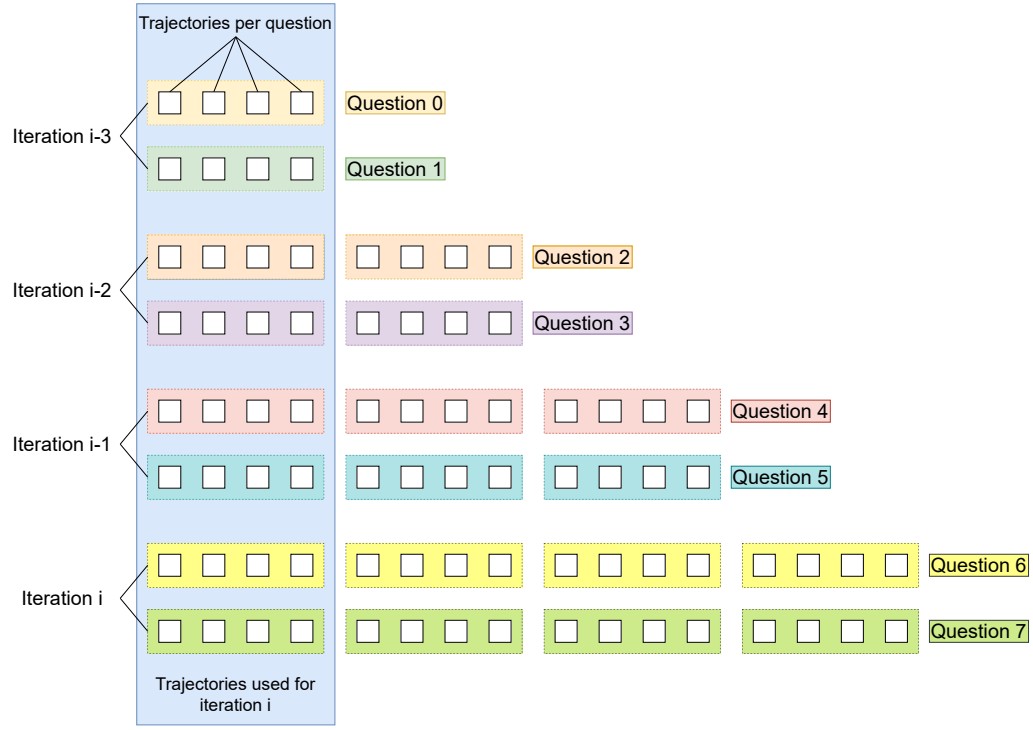

Figure 7: Illustration of the replay buffer. In this example, in each iteration, we sample two questions, with each question generating four trajectories. These trajectories are distributed across four iterations for optimization. The trajectories used in the $i$-th iteration are selected in the blue box. In total, there are 8 different questions, each with 4 trajectories, ensuring a balanced distribution of samples.

## D   Replay Buffer

Within each iteration, sampling a large number of trajectories from a single question can lead to imbalanced training, causing the model to overfit to specific samples. To address this issue, we introduce a replay buffer mechanism that distributes sampled trajectories across multiple iterations, ensuring a more balanced question distribution, as shown in Figure 7. Additionally, PPO's clip mechanism helps maintain stable optimization when using these trajectories from previous iterations. Specifically, in our experiments, we use a *6-6-6* tree structure for sampling, generating $6 \times 6 \times 6 = 216$ trajectories per question. In each iteration, we sample 16 distinct questions to generate trajectories, and these trajectories are then distributed across the following 8 iterations, with at most 32 trajectories per question processed in each iteration. As a result, each iteration covers 128 distinct questions (8 × 16), ensuring a well-balanced sample distribution. We observe that the clip ratio remains around 0.1, validating the effectiveness of our replay buffer strategy.

Our replay buffer mechanism leads to a certain degree of off-policy learning, enabling our framework to be compatible with the asynchronous RL setting, which was first explored by [20]. Leveraging asynchronous RL can further parallelize data collection and policy optimization, potentially leading to additional efficiency improvements.

## E   Pseudo-code of Tree-Structured Advantage Estimation Method

We efficiently construct the tree using Python's asyncio library. For the detailed algorithm procedure, please refer to Algorithm 1 below.

**Algorithm 1** Tree-structured Advantage Estimation Method

1: **procedure** CONSTRUCTTREE(*prompt*, *max_depth*, *structure*, *M*, *instance*, *adv_method*)
2:     **create root node:**
3:         $root.text \leftarrow prompt$
4:         $root.full\_text \leftarrow prompt$
5:         $root.depth \leftarrow 0$
6:     **await** BUILD(*root*, *prompt*, 0, *max_depth*, *structure*, *M*, *instance*)
7:     COMPUTEADVANTAGE(*root*, *None*, *adv_method*)
8:     **return** *root*
9: **end procedure**
10: **procedure** BUILD(*node*, *prefix*, *depth*, *max_depth*, *structure*, *M*, *instance*)
11:     **if** $depth \geq max\_depth$ **then**
12:         $node.reward \leftarrow$ REWARD_FUNCTION(*prefix*, *node.text*, *instance*)
13:         **return**
14:     **end if**
15:     $K \leftarrow structure[depth]$
16:     **if** $depth < max\_depth - 1$ **then**
17:         $max\_tokens \leftarrow M$
18:     **else**
19:         $max\_tokens \leftarrow$ None
20:     **end if**
21:     $children \leftarrow$ **await** EXPAND(*prefix*, *depth*, *K*, *max_tokens*)
22:     $node.children \leftarrow children$
23:     initialize empty list *expansion_tasks*
24:     **for** each *child* in *children* **do**
25:         **if** $child.finish\_reason \neq$ "length" **then**
26:             $child.reward \leftarrow$ REWARD_FUNCTION(*prefix*, *child.text*, *instance*)
27:         **else**
28:             create async task $t \leftarrow$ BUILD(*child*, *child.full_text*, $depth + 1$, *max_depth*,
                      *structure*, *M*, *instance*)
29:             append *t* to *expansion_tasks*
30:         **end if**
31:     **end for**
32:     **await** GATHER(*expansion_tasks*)
33:     $node.reward \leftarrow$ mean(rewards of *node.children*)
34:     $node.reward\_std \leftarrow$ std(rewards of *node.children*)
35: **end procedure**
36: **procedure** EXPAND(*prefix*, *depth*, *K*, *max_tokens*)
37:     construct LLM prompt from template using *prefix*
38:     create async task calling LLM API requesting $K$ responses, each limited to *max_tokens*
39:     $responses \leftarrow$ **await** *task*
40:     initialize empty list *children*
41:     **for** each *response* in *responses* **do**
42:         **create new node** *child*:
43:             $child.text \leftarrow response.text$
44:             $child.finish\_reason \leftarrow response.finish\_reason$
45:             $child.full\_text \leftarrow$ concatenate_texts(*prefix*, *child.text*)
46:         append *child* to *children*
47:     **end for**
48:     **return** *children*
49: **end procedure**
50: **procedure** COMPUTEADVANTAGE(*node*, *parent*, *adv_method*)
51:     **if** *parent* is not *None* **then**
52:         **if** $adv\_method =$ "RLOO" **then**
53:             $node.advantage \leftarrow node.reward - parent.reward$
54:         **else if** $adv\_method =$ "GRPO" **then**
55:             $node.advantage \leftarrow (node.reward - parent.reward)/parent.reward\_std$
56:         **end if**
57:     **end if**
58: **end procedure**

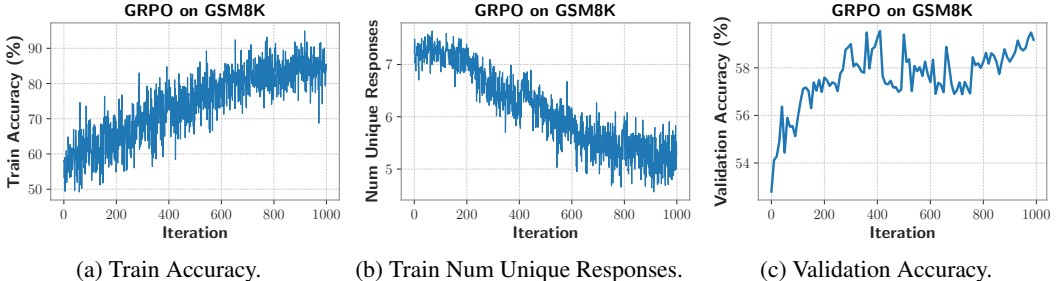

(a) Train Accuracy.  (b) Train Num Unique Responses.  (c) Validation Accuracy.

Figure 8: GRPO on GSM8K exhibits rapid overfitting: while training accuracy improves steadily (Fig. 8a), but the unique response number constantly drops (Fig. 8b), and validation accuracy saturates early (Fig. 8c).

# F   Additional Experiment Results

We also evaluate our SPO-chain method using DeepSeekMath 7B on the MATH dataset, as shown in Figure 9. Our approach achieves performance comparable to VinePPO and surpasses all other baselines. Due to computational constraints, we initially train with an interval of 10 and later switch to 5. After switching from the interval of 10 to 5, the model performance continues to improve, suggesting that using a finer-grained advantage is beneficial for model training. Notably, as shown in Figure 10, our method uses significantly fewer segments than VinePPO while attaining similar performance.

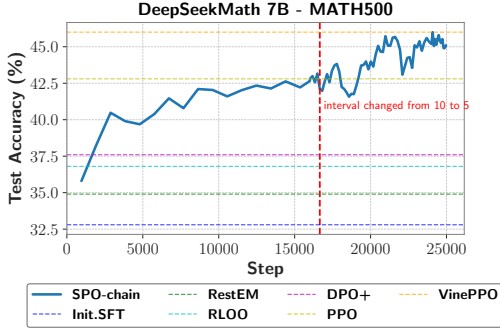

Figure 9: Performance of SPO-chain on DeepSeekMath 7B evaluated on the MATH500. Baseline results are from [12].

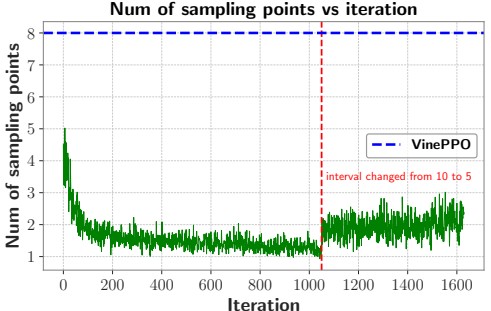

Figure 10: Comparison of the number sampling points between our method and VinePPO. According to [9], there are approximately 8 reasoning steps on average for MATH.

In addition, We conduct experiments on instruct and base models. Specifically, we evaluate Qwen2.5-1.5B-Instruct and Qwen2.5-0.5B-Instruct on the GSM8K dataset, and Qwen2.5-Math-1.5B on the MATH dataset. The results are presented in Figure 11 (a)–(c). Across all these settings, our method consistently outperforms GRPO, achieving higher test accuracy.

We further extend our evaluation beyond mathematical reasoning to the Knights-and-Knaves dataset [33], a classic logic puzzle benchmark where each character is either a knight (who always tells the truth) or a knave (who always lies), and the task is to deduce each character's identity from their statements. For this experiment, we use the 3-people (3ppl) subset and train models based on Qwen2.5-1.5B-Instruct. The corresponding results are reported in Figure 11 (d). On this dataset as well, our method outperforms GRPO, demonstrating its generality and effectiveness beyond the MATH dataset.

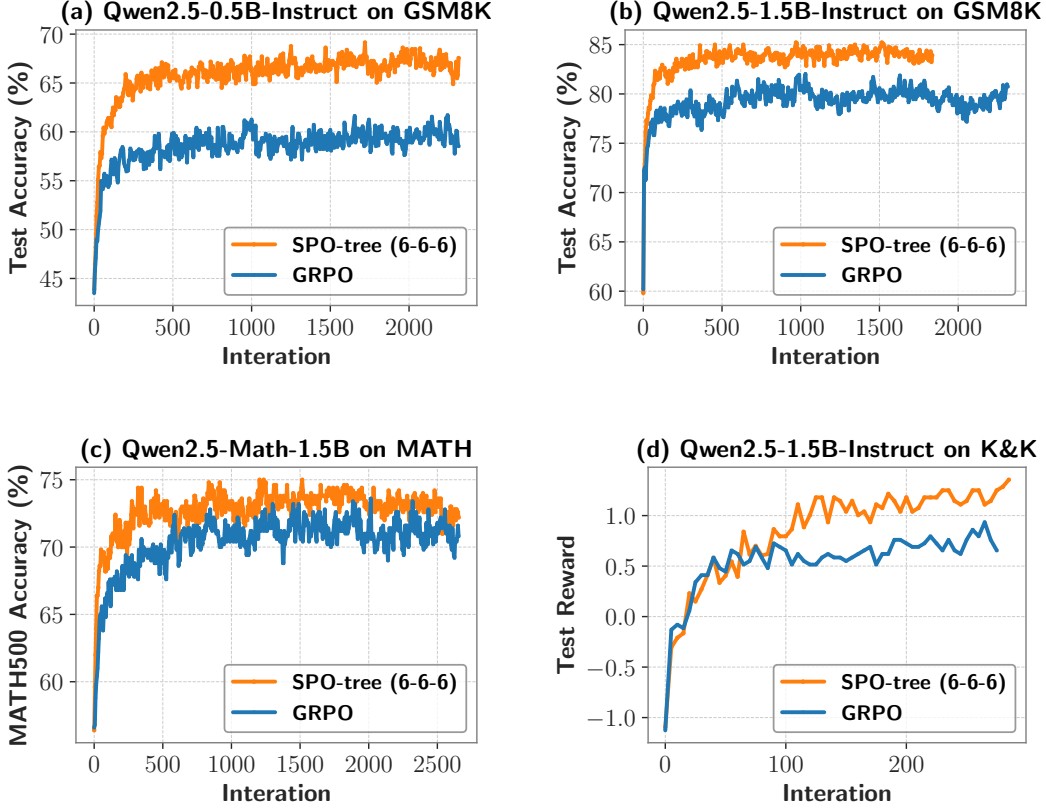

Figure 11: Experimental results across different models and datasets. (a) Qwen2.5-0.5B-Instruct on GSM8K. (b) Qwen2.5-1.5B-Instruct on GSM8K. (c) Qwen2.5-1.5B-Math on MATH. (d) Qwen2.5-1.5B-Instruct on Knights-and-Knaves (3ppl).

## G   Limitations and Future Work

In long CoT scenario, we propose a tree-based advantage estimation strategy that can deliver fine-grained reward signals with lower sampling costs. However, for a single problem, constructing a tree requires sampling a large number of trajectories. We use a replay buffer to distribute these trajectories across several future iterations. However, this approach is still constrained by on-policy algorithms, which cannot distribute trajectories too far into future iterations. In the future, we may explore more off-policy algorithms, to achieve more efficient sample utilization. This would allow us to reuse the high-quality samples obtained from SPO-chain methods and alleviate the issue of sample imbalance caused by tree-based sampling.

## H   Hyperparameters and Compute Resources

Our code base is based on [12]. Most hyperparameters are the same as theirs. Here we list important hyperparameters in the following five tables. We perform SPO-chain and SPO-tree experiments with RhoMath 1.1B using a single A100 GPU (40GB). For the long-CoT experiments (2K and 4K context) using DeepSeek-R1-Distill-Qwen-1.5B, we use a single A100 GPU (80GB).

**SPO-chain (int5) Rho 1.1B on GSM8K**

| Hyperparameter | Value |
| --- | --- |
| Target train batch size | 64 |
| Num episodes per iteration | 512 |
| Dataset samples per iteration | 64 |
| Samples (per prompt) | 8 |
| Cutpoint interval | 5 |
| K (MC samples) | 9 |
| M (Tokens per level) | - |
| Branch factors | - |
| Train policy temperature | 0.6 |
| Train policy top-p | 0.9 |
| Train context size | 2047 |
| Initial KL coef | 0.0001 |
| Learning rate | $1 \times 10^{-6}$ |
| Epochs per iteration | 2 |
| Prob mask threshold | 0.9 |
| Eval. context size | 2047 |
| Eval. temperature | 0.35 |
| Eval. top-p | 0.9 |
| Eval. samples | 16 |

**SPO-tree (6-6-6) Rho 1.1B on GSM8K**

| Hyperparameter | Value |
| --- | --- |
| Target train batch size | 128 |
| Num episodes per iteration | 512 |
| Dataset samples per iteration | 16 |
| Samples (per prompt) | - |
| Cutpoint interval | - |
| K (MC samples) | - |
| M (Tokens per level) | 30 |
| Branch factors | [6,6,6] |
| Train policy temperature | 0.6 |
| Train policy top-p | 0.9 |
| Train context size | 2047 |
| Initial KL coef | 0.0001 |
| Learning rate | $1 \times 10^{-6}$ |
| Epochs per iteration | 1 |
| Prob mask threshold | 0.9 |
| Eval. context size | 2047 |
| Eval. temperature | 0.35 |
| Eval. top-p | 0.9 |
| Eval. samples | 16 |

**GRPO DeepSeek-R1-Distill-Qwen 1.5B on MATH**

| Hyperparameter | Value |
| --- | --- |
| Target train batch size | 128 |
| Num episodes per iteration | 512 |
| Dataset samples per iteration | 64 |
| Samples (per prompt) | 8 |
| Cutpoint interval | - |
| K (MC samples) | - |
| M (Tokens per level) | - |
| Branch factors | - |
| Train policy temperature | 0.6 |
| Train policy top-p | 1 |
| Train context size | 2048 => 4096 |
| Initial KL coef | 0.0001 |
| Learning rate | $1 \times 10^{-6}$ |
| Epochs per iteration | 1 |
| Prob mask threshold | No prob mask |
| Eval. context size | 2048 => 4096 |
| Eval. temperature | 0 |
| Eval. top-p | 0.9 |
| Eval. samples | 1 |

**SPO-tree (6-6-6) DeepSeek-R1-Distill-Qwen 1.5B on MATH**

| Hyperparameter | Value |
| --- | --- |
| Target train batch size | 128 |
| Num episodes per iteration | 1024 |
| Dataset samples per iteration | 16 |
| Samples (per prompt) | - |
| Cutpoint interval | - |
| K (MC samples) | - |
| M (Tokens per level) | 600 |
| Branch factors | [6,6,6] |
| Train policy temperature | 0.6 |
| Train policy top-p | 1 |
| Train context size | 2048 => 4096 |
| Initial KL coef | 0.0001 |
| Learning rate | $1 \times 10^{-6}$ |
| Epochs per iteration | 1 |
| Prob mask threshold | 0.9 |
| Eval. context size | 2048 => 4096 |
| Eval. temperature | 0 |
| Eval. top-p | 0.9 |
| Eval. samples | 1 |

**SPO-chain (int5=>10) DeepSeekMath 7B on MATH**

| Hyperparameter | Value |
| --- | --- |
| Target train batch size | 64 |
| Num episodes per iteration | 512 |
| Dataset samples per iteration | 64 |
| Samples (per prompt) | 8 |
| Cutpoint interval | 10 => 5 |
| K (MC samples) | 9 |
| M (Tokens per level) | - |
| Branch factors | - |
| Train policy temperature | 0.6 |
| Train policy top-p | 0.9 |
| Train context size | 4095 |
| Initial KL coef | 0.0001 |
| Learning rate | $1 \times 10^{-6}$ |
| Epochs per iteration | 2 |
| Prob mask threshold | 0.9 |
| Eval. context size | 4095 |
| Eval. temperature | 0.35 |
| Eval. top-p | 0.9 |
| Eval. samples | 16 |

# I   Segment Partition Example

**Prompt:** [MATH_TASK] Problem: Gloria wants to buy the $129,000 mountain cabin that her friend Alfonso is selling. She only has $150 in cash. She intends to raise the remaining amount by selling her mature trees for lumber. She has 20 cypress trees, 600 pine trees, and 24 maple trees. She will get $100 for each cypress tree, $300 for a maple tree, and $200 per pine tree. After paying Alfonso for the cabin, how much money will Gloria have left? Solution:

The total value of the cypress trees that she will | sell is⬚20 * $100 = $2000 .

The total value of the maple trees that she will sell is  | 600 * $300 = $18000.

The total value of the pine trees that she will sell is 24 * $200 = $4800.

The total value of all the trees that she will sell is $2000 + $18000 + $4800 = $2 4800.

| After selling all the trees, she will have $24800 - $150 = $23300 left.

#### 23300

Figure 12: Segment partition based on fixed *cutpoint* interval. Each row corresponds to a single step. Tokens marked in red indicate cutpoints (tokens with probability lower than 0.9). The blue vertical lines represent SPO segment boundaries.

Figure 12 shows a practical segment partition example. As we can see, many of the cutpoints (tokens whose probability is below a threshold) correspond to the places where the model makes mistakes. For instance, in the second step, the total value of the maple trees should be 24 * $300, but the model outputs 600 * 300 and shows low confidence at the digit 6. If it had output 2 in this position, this step might have been correct. Similarly, in the final step, the calculation should be 24800 - 12900, but the model instead outputs 24800 - 150 and is uncertain at the digit 5. Had it produced a 2 at this point, the equation might have been correct. This case shows that cutpoints are the locations where the model's reasoning trajectory can shift, and they are the main drivers behind "segment advantage". Therefore, identifying and accurately assigning credit to these cutpoints is essential, which is consistent with our segment partition method (Section 4) and probability mask technique (Section 4).

It is also worth noting that partitioning by line breaks, as in VinePPO, is not the most effective approach. For example, in step three, none of the tokens have probabilities below 0.9, indicating that the transition probabilities for all tokens in this step are very high. As a result, the estimated $V$ value at the beginning and end of this step will not differ significantly, thus the advantage for this segment will be close to zero. Allocating sampling budget to this step is likely to result in wasted samples.

Moreover, partitioning by step boundaries can lead to an uneven distribution of cutpoints, i.e., some segments may contain too many, while others include too few. Our goal in segment partition is to have each segment contain roughly the same number of potential "turning points", so that we can assign credit as efficiently as possible within a limited sampling budget. Step-based partition does not achieve this balance. This may help explain why our SPO-chain method can outperform VinePPO, even with fewer sampling points.

## J  Model Output Example

Compared to DeepSeek-R1-Distill-Qwen-1.5B, our optimized model demonstrates greater token efficiency, requiring fewer tokens to arrive at the correct answer (both models use greedy decoding). When the model's context length is limited, increased token efficiency significantly enhances performance.

---

### Problem

What is the smallest positive integer $n$ such that $17n \equiv 1234 \pmod 7$?

---

### SPO-tree

Okay, so I need to find the smallest positive integer $n$ such that $17n \equiv 1234 \pmod 7$. Hmm, let me think about how to approach this. I remember that when dealing with congruences modulo a number, it's often helpful to simplify the coefficients modulo that number first. So, maybe I should start by reducing 17 and 1234 modulo 7.

Let me recall that to find a number modulo 7, I can divide by 7 and find the remainder. So, let's start with 17. Dividing 17 by 7, 7 goes into 17 twice with a remainder. 2 times 7 is 14, and 17 minus 14 is 3. So, 17 is congruent to 3 modulo 7. That means $17 \equiv 3 \pmod 7$. So, the equation $17n \equiv 1234 \pmod 7$ simplifies to $3n \equiv 1234 \pmod 7$.

Now, I need to find 1234 modulo 7. Let me do that. 1234 divided by 7. Hmm, 7 times 176 is 1232 because 176 times 7 is 1232. So, 1234 minus 1232 is 2. Therefore, $1234 \equiv 2 \pmod 7$. So, now the equation is $3n \equiv 2 \pmod 7$.

Alright, so I have $3n \equiv 2 \pmod 7$. I need to solve for $n$. To do that, I think I need to find the multiplicative inverse of 3 modulo 7. The multiplicative inverse of 3 is a number $x$ such that $3x \equiv 1 \pmod 7$. Once I have that, I can multiply both sides of the equation by $x$ to get $n \equiv 2x \pmod 7$.

So, let me find the inverse of 3 modulo 7. Let's try numbers from 1 upwards:
- $3 \times 1 = 3 \equiv 3 \pmod 7 \to$ Not 1. - $3 \times 2 = 6 \equiv 6 \pmod 7 \to$ Not 1. - $3 \times 3 = 9 \equiv 2 \pmod 7 \to$ Not 1. - $3 \times 4 = 12 \equiv 5 \pmod 7 \to$ Not 1. - $3 \times 5 = 15 \equiv 1 \pmod 7 \to$ Yes! So, 5 is the inverse because 3*5=15, which is 1 mod 7.

Great, so the inverse of 3 mod 7 is 5. Therefore, multiplying both sides of $3n \equiv 2 \pmod 7$ by 5 gives:
$5 \times 3n \equiv 5 \times 2 \pmod 7$
Simplifying, $15n \equiv 10 \pmod 7$. But 15 mod 7 is 1, because 7*2=14, so 15-14=1. Similarly, 10 mod 7 is 3, because 7*1=7, 10-7=3. So, this simplifies to:
$1n \equiv 3 \pmod 7$
Which means $n \equiv 3 \pmod 7$. So, the smallest positive integer $n$ that satisfies this is 3. Let me just verify that.

Calculating $17 * 3 = 51$. Then, 51 mod 7: 7*7=49, so 51-49=2. And 1234 mod 7 is 2, as we found earlier. So, 51 mod 7 is 2, which matches 1234 mod 7. So, yes, 3 is the correct answer.

**Final Answer** The smallest positive integer $n$ is $\boxed{3}$.
To find the smallest positive integer $n$ such that $17n \equiv 1234 \pmod 7$, we start by simplifying the coefficients modulo 7.

First, we reduce 17 modulo 7:

$$17 \div 7 = 2 \text{ remainder } 3 \implies 17 \equiv 3 \pmod 7$$

Thus, the equation $17n \equiv 1234 \pmod 7$ simplifies to:

$$3n \equiv 1234 \pmod 7$$

Next, we reduce 1234 modulo 7:

$$1234 \div 7 = 176 \text{ remainder } 2 \implies 1234 \equiv 2 \pmod 7$$

So, the equation becomes:

$$3n \equiv 2 \pmod 7$$

To solve for $n$, we need the multiplicative inverse of 3 modulo 7. Testing values, we find that:

$$3 \times 5 = 15 \equiv 1 \pmod 7$$

Thus, the inverse of 3 modulo 7 is 5. Multiplying both sides of $3n \equiv 2 \pmod 7$ by 5:

$$5 \times 3n \equiv 5 \times 2 \pmod 7 \implies 15n \equiv 10 \pmod 7$$

Simplifying, since $15 \equiv 1 \pmod 7$ and $10 \equiv 3 \pmod 7$:

$$n \equiv 3 \pmod 7$$

The smallest positive integer $n$ is 3. Verifying, $17 \times 3 = 51 \equiv 2 \pmod 7$ and $1234 \equiv 2 \pmod 7$, confirming the solution.

$$\boxed{3}$$

## DeepSeek-R1-Distill-Qwen-1.5B

Okay, so I have this problem here: find the smallest positive integer $n$ such that $17n \equiv 1234 \pmod 7$. Hmm, modular arithmetic, right? I remember that congruences modulo 7 mean that when you divide $17n - 1234$ by 7, there's no remainder. So, basically, $17n$ and $1234$ leave the same remainder when divided by 7.

First, maybe I should simplify both sides of the congruence modulo 7. That is, find what 17 is equivalent to modulo 7, and what 1234 is equivalent to modulo 7. That should make the equation easier to handle.

Starting with 17 modulo 7. Let me divide 17 by 7. 7 goes into 17 twice, which is 14, and the remainder is 3. So, $17 \equiv 3 \pmod 7$. That simplifies the left side of the congruence to $3n$.

Now, the right side is 1234 modulo 7. Hmm, 1234 is a big number, so I need a systematic way to find its remainder when divided by 7. Maybe I can break it down digit by digit or use some divisibility rules. Alternatively, I can use the fact that 10 is congruent to 3 modulo 7, so powers of 10 can be simplified.

Wait, another approach: I can divide 1234 by 7 and find the remainder. Let me do that. 7 goes into 12 once, with a remainder of 5. Bring down the 3 to make 53. 7 goes into 53 seven times, which is 49, leaving a remainder of 4. Bring down the 4 to make 44. 7 goes into 44 six times, which is 42, leaving a remainder of 2. So, 1234 divided by 7 is 176 with a remainder of 2. Therefore, $1234 \equiv 2 \pmod 7$.

So now, the congruence simplifies to $3n \equiv 2p \mod 7$. Okay, so I need to solve for $n$ in this equation. That is, find the smallest positive integer $n$ such that when multiplied by 3 and then divided by 7, the remainder is 2.

To solve $3n \equiv 2 \pmod 7$, I think I need to find the multiplicative inverse of 3 modulo 7. The multiplicative inverse of 3 is a number $x$ such that $3x \equiv 1 \pmod 7$. Once I have that, I can multiply both sides of the congruence by $x$ to isolate $n$.

So, let's find the inverse of 3 modulo 7. I can test numbers from 1 upwards to see which one satisfies $3x \equiv 1 \pmod 7$.

- $3 \times 1 = 3 \equiv 3 \pmod 7 \rightarrow$ Not 1. - $3 \times 2 = 6 \equiv 6 \pmod 7 \rightarrow$ Not 1. - $3 \times 3 = 9 \equiv 2 \pmod 7 \rightarrow$ Not 1. - $3 \times 4 = 12 \equiv 5 \pmod 7 \rightarrow$ Not 1. - $3 \times 5 = 15 \equiv 1 \pmod 7 \rightarrow$ Yes! That works.

So, the inverse of 3 modulo 7 is 5. Therefore, multiplying both sides of the congruence $3n \equiv 2 \pmod 7$ by 5 gives:

$5 \times 3n \equiv 5 \times 2 \pmod 7$

Simplifying the left side: $15n \equiv 2 \pmod 7$. But 15 modulo 7 is 1, since 15 divided by 7 is 2 with a remainder of 1. So, this simplifies to:

$1n \equiv 2 \pmod{7}$, which is just $n \equiv 2 \pmod{7}$.

Therefore, the smallest positive integer $n$ that satisfies this congruence is 2. Let me double-check to make sure I didn't make a mistake.

Plugging $n = 2$ back into the original equation: $17 \times 2 = 34$. Now, 34 divided by 7 is 4 with a remainder of 6. Wait, but 1234 modulo 7 was 2, so 34 modulo 7 is 6, which is not equal to 2. Hmm, that's a problem. Did I do something wrong?

Wait, hold on. Let me check my steps again. Maybe I messed up the inverse or the simplification.

Starting over: $17n \equiv 1234 \pmod{7}$. I found that $17 \equiv 3 \pmod{7}$ and $1234 \equiv 2 \pmod{7}$. So, $3n \equiv 2 \pmod{7}$. Then, I found the inverse of 3 modulo 7 is 5 because $3 \times 5 = 15 \equiv 1 \pmod{7}$. So, multiplying both sides by 5: $n \equiv 10 \pmod{7}$. But 10 modulo 7 is 3, so $n \equiv 3 \pmod{7}$. Wait, so earlier I thought it was 2, but that was a mistake.

Wait, no, let me recast that. If $n \equiv 10 \pmod{7}$, then 10 divided by 7 is 1 with a remainder of 3, so $n \equiv 3 \pmod{7}$. So, the smallest positive integer $n$ is 3. Let me check that.

Plugging $n = 3$ into $17n$: $17 \times 3 = 51$. 51 divided by 7 is 7 with a remainder of 2. So, 51 modulo 7 is 2, which matches the right side of the congruence. So, that works.

Wait, so earlier I thought the inverse was 5, but that led me to $n \equiv 10 \pmod{7}$, which is 3. So, the correct answer is 3, not 2. I must have made a mistake in my initial calculation.

Let me go through the inverse part again. So, to find the inverse of 3 modulo 7, I need a number $x$ such that $3x \equiv 1 \pmod{7}$. Testing:

- $3 \times 1 = 3 \equiv 3 \pmod{7}$ - $3 \times 2 = 6 \equiv 6 \pmod{7}$ - $3 \times 3 = 9 \equiv 2 \pmod{7}$ - $3 \times 4 = 12 \equiv 5 \pmod{7}$ - $3 \times 5 = 15 \equiv 1 \pmod{7}$

Yes, so 5 is the inverse. So, multiplying both sides by 5: $5 \times 3n \equiv 5 \times 2 \pmod{7}$, which is $15n \equiv 10 \pmod{7}$. But 15 modulo 7 is 1, so $n \equiv 10 \pmod{7}$. 10 divided by 7 is 1 with a remainder of 3, so $n \equiv 3 \pmod{7}$. So, the smallest positive integer is 3.

Wait, so why did I get confused earlier? Because I thought 10 modulo 7 is 3, but I thought 10 was 3, but when I first thought of 10, I thought it was 3, but when I first thought of 10, I thought it was 3, but when I thought of 10, I thought it was 3, but when I thought of 10, I thought it was 3, but when I thought of 10, I thought it was 3, but when I thought of 10, I thought it was 3. Wait, no, that's not right.

Wait, 10 divided by 7 is 1 with a remainder of 3, so 10 modulo 7 is 3. So, $n \equiv 3 \pmod{7}$. So, the smallest positive integer is 3.

Let me verify again: $17 \times 3 = 51$. 51 divided by 7 is 7 with a remainder of 2. So, 51 modulo 7 is 2, which is equal to 1234 modulo 7, which is 2. So, that works.

Wait, so earlier, when I thought I had 10 modulo 7, I thought it was 3, but I think I confused myself. So, the correct answer is 3.

But just to make sure, let me try $n = 3$: $17 \times 3 = 51$. 51 divided by 7 is 7*7=49, so 51-49=2. So, 51 mod7=2, which is equal to 1234 mod7=2. So, yes, that works.

Wait, but earlier, when I thought of 10 modulo7, I thought it was 3, but that was a miscalculation. So, the correct inverse is 5, leading to $n \equiv 10 \pmod{7}$, which is 3.

So, the smallest positive integer is 3.

But just to make sure, let me try $n = 3$ in the original equation: $17 \times 3 = 51$. 51 mod7 is 2, which is equal to 1234 mod7, which is 2. So, yes, that works.

Wait, but let me also check $n = 3 - 7 = -4$. But since we're looking for positive integers, 3 is the smallest.

Alternatively, if I had a different inverse, but no, 5 is the inverse.

Wait, but just to make sure, let me try $n = 3 + 7 = 10$. $17 \times 10 = 170$. 170 divided by 7 is 24*7=168, so 170-168=2. So, 170 mod7=2, which is equal to 1234 mod7=2. So, that also works, but 10 is larger than 3, so 3 is the smallest.

Therefore, the answer is 3.

Wait, but hold on, in my initial calculation, I thought I had 10 modulo7=3, but I thought 10 was 3, but that was a confusion. So, the correct answer is 3.

But just to make sure, let me try another approach. Maybe I can solve $3n \equiv 2 \pmod{7}$ by testing n=1,2,3,... until I find the smallest n that satisfies the equation.

n=1: $3 * 1 = 3 \equiv 3 \neq 2 \pmod{7}$ n=2: $3 * 2 = 6 \equiv 6 \neq 2 \pmod{7}$ n=3: $3 * 3 = 9 \equiv 2 \pmod{7}$. Yes! So, n=3 is the smallest positive integer.

So, that confirms it. So, the answer is 3.

Wait, but in my initial thought process, I thought I had 10 modulo7=3, but that was a miscalculation. So, the correct answer is 3.

Therefore, the smallest positive integer n is 3.

**Final Answer** The smallest positive integer $n$ is $\boxed{3}$.

To find the smallest positive integer $n$ such that $17n \equiv 1234 \pmod{7}$, we start by simplifying both sides of the congruence modulo 7.

First, we simplify $17 \mod 7$:

$$17 \div 7 = 2 \text{ remainder } 3$$

Thus, $17 \equiv 3 \pmod{7}$.

Next, we simplify $1234 \mod 7$. We can do this by dividing 1234 by 7 and finding the remainder:

$$1234 \div 7 = 176 \text{ remainder } 2$$

Thus, $1234 \equiv 2 \pmod{7}$.

This reduces our original congruence to:

$$3n \equiv 2 \pmod{7}$$

To solve for $n$, we need the multiplicative inverse of 3 modulo7 . We find this by testing values:

$$3 \times 5 = 15 \equiv 1 \pmod{7}$$

Thus, the inverse of 3 modulo 7 is 5.

Multiplying both sides of the congruence $3n \equiv 2 \pmod{7}$ by 5:

$$5 \times 3n \equiv 5 \times 2 \pmod{7}$$

$$15n \equiv 10 \pmod{7}$$

Since $15 \equiv 1 \pmod{7}$, this simplifies to:

$$n \equiv 10 \pmod{7}$$

$$10 \div 7 = 1 \text{ remainder } 3$$

Thus, $n \equiv 3 \pmod{7}$.

The smallest positive integer $n$ is 3. Verifying, we find:

$$17 \times 3 = 51$$

$$51 \div 7 = 7 \text{ remainder } 2$$

Thus, $51 \equiv 2 \pmod{7}$, confirming our solution.

The smallest positive integer $n$ is $\boxed{3}$.

