# OpenReview forum: "Segment Policy Optimization: Effective Segment-Level Credit Assignment in RL for Large Language Models"
_NeurIPS.cc/2025/Conference — NeurIPS 2025 poster_

### Official Review · Reviewer_L77S · 2025-06-22

**Clarity:** 3
**Significance:** 3
**Originality:** 3
**Rating:** 5
**Confidence:** 3

**Summary:**

The authors propose a new algorithm for training reasoning models with reinforcement learning by improving credit assignment. Token-level methods like PPO has a value network, which is expensive to calculate and can be inaccurate, whereas trajectory-level methods like GRPO is too crude when it comes to assigning credits.

To help bridge the gap, the authors propose a new method called Segment Policy Optimization (SPO), breaking the trajectory into segments. If the trajectory is typically short, SPO performs the segmentation mostly by identifying token positions where the probabilities drop below a certain threshold, which usually corresponds to where the reasoning trajectory might diverge. These positions are identified as "cutpoints". Then they can just perform Monte Carlo on those cutpoints to do value estimation. For longer trajectories, SPO employs a tree based method that attempts to re-use existing calculations.

The experiments show that SPO methods can outperform many other RL methods like GRPO and PPO when evaluated using the gsm8k test accuracy, when trained on RhoMath 1.1B.

**Questions:**

* Could the authors provide more runtime or compute budget information on SPO? For example, using interval 5 seems to use a lot more computation than interval 100. However, this is not quantified in Figure 4.

* Can you plot the learning curves using time as the x-axis?

**Ethical Concerns:**

["NO or VERY MINOR ethics concerns only"]

**Final Justification:**

The author provided more information on runtime and cutpoint analysis, which answers my question. The paper is well-written and it contributes a new idea for reasoning research. I am raising my score because of this.

**Quality:**

3

**Strengths And Weaknesses:**

Strengths:

The paper is well-motivated; the authors compare several methods for credit assignment on trajectory segments. Furthermore, the authors came up with clever heuristics like determining segment boundaries via "cutpoints", where the probability of predicting the next token drops.

Experiments seem well-designed, and ablation studies were provided.

Weaknesses:

While motivated, I'd like to see more visual examples. For example, the authors identify the cutpoints. I wonder if they could share actual examples of these cutpoints. I also wonder if these cutpoints always correspond to the end of a reasoning step, or could they be something else entirely.

I'd also like to see more runtime / compute information being provided for SPO. For example, in Figure (4), interval 5 seems a lot more expensive to run than interval 100. However, this is not quantified, and in a way it's not a fair comparison between interval 5 and interval 100. One char that's useful is using the compute units (e.g., FLOPS or a crude runtime) on the x-axis, and the accuracy on the y-axis, using a scatter plot.

---

> ### Author Rebuttal · Authors · 2025-07-31
>
> Thanks for your valuable and constructive comments. Below, we provide detailed responses to the concerns and questions you have raised.
>
> > W1: While motivated, I'd like to see more visual examples. For example, the authors identify the cutpoints. I wonder if they could share actual examples of these cutpoints. I also wonder if these cutpoints always correspond to the end of a reasoning step, or could they be something else entirely.
>
> In the previously submitted Appendix, we provided examples of cutpoints in Appendix I—please see Figure 11, where the red tokens indicate cutpoints. Many cutpoints correspond to numbers in the equations, and there are also cases where a cutpoint occurs at the beginning of a reasoning step or within a reasoning step.
>
> >W2: I'd also like to see more runtime / compute information being provided for SPO. For example, in Figure (4), interval 5 seems a lot more expensive to run than interval 100. However, this is not quantified, and in a way it's not a fair comparison between interval 5 and interval 100. One char that's useful is using the compute units (e.g., FLOPS or a crude runtime) on the x-axis, and the accuracy on the y-axis, using a scatter plot. Could the authors provide more runtime or compute budget information on SPO? For example, using an interval 5 seems to use a lot more computation than an interval 100. However, this is not quantified in Figure 4.
>
> Thank you for your suggestion regarding runtime and compute information.
>
> For the experiments in Figure 4(a), we measured the average wall-clock time per 10 training iterations under different interval settings:
>
> - **Interval 2 (int2):** 1.43 hours per 10 iterations
> - **Interval 5 (int5):** 0.87 hours per 10 iterations
> - **Interval 100 (int100):** 0.58 hours per 10 iterations
>
> We also measured the validation accuracy against the training wall-clock time for the three settings, as shown in the following table:
>
> | | | | |
> |:-:|:-:|:-:|:-:|
> |Training hours / Accuracy (%)|int2|int5|int100|
> |10|58.1|**62.7**|57.5|
> |20|61.3|**65.7**|60.9|
> |30|64.3|**67.9**|64.0|
> |40|66.4|**69.4**|64.3|
> |50|67.6|**70.7**|64.6|
> |60|68.6|**70.7**|64.6|
> |70|69.1|**70.7**|64.6|
> |80|69.7|**70.7**|64.6|
> |90|70.4|**70.7**|64.6|
> |100|70.4|**70.7**|64.6|
> |110|**70.8**|70.7|64.6|
> |120|**71.7**|70.7|64.6|
>
> The results show that, within the first 100 hours of training, interval 5 achieves the highest validation accuracy, followed by interval 2, with interval 100 being the lowest. In terms of final accuracy, interval 2 achieves slightly higher accuracy than interval 5, while interval 100 trails further behind. Overall, these results indicate that using a moderate interval (such as interval 5) offers a more efficient trade-off, achieving better validation accuracy given the same training time compared to very short or very long intervals.

---

> ### Comment · Reviewer_L77S · 2025-08-01
> **Reply**
>
> Thanks for the detailed reply. I appreciate the example on cutpoints and wall clock time.

---

### Official Review · Reviewer_vcBi · 2025-07-02

**Clarity:** 2
**Significance:** 2
**Originality:** 3
**Rating:** 3
**Confidence:** 3

**Summary:**

This work innovatively proposes to shift from the calculation of fine-grained token-level advantages to a coarse-grained advantage calculation at the sub-trajectory (fragment) level. As a result, it achieves a better balance between credibility allocation and computational efficiency compared to VinePPO. It provides more accurate credit allocation than the trajectory-level method and requires fewer estimation points than the token-level method, enabling advantage estimation based on the Monte Carlo (MC) method without the need for an evaluation model. The method has been validated for its effectiveness on two simple and moderately difficult datasets.

**Questions:**

Apart from what has been mentioned above, I noticed that the introduction of the experimental part seems to lack crucial information. 1. As VinePPO is the main comparison method, did it also set the same number of MC rollout to ensure the fairness of the experiment? 2. The basic model is too simple and has already been distilled using R1's mathematical data in advance. This makes the base model itself already have very good scores in mathematics (as shown in the author's experimental comparison table). The author should conduct experiments and verification in 1.5-base or qwen-1.6b-base, align with other RL-zero papers. (3) It seems that the actual response generation of LLM has not been released.

**Ethical Concerns:**

["NO or VERY MINOR ethics concerns only"]

**Final Justification:**

Thanks to the author for the reply, I'll keep my score

**Limitations:**

See above

**Quality:**

3

**Strengths And Weaknesses:**

Strength: The paper is well-written, and the idea is interesting to me. The experimental results are good with clear visualization.

Weakness: The design details of the method are rather detailed, but the design motivation is too vague and empirical. It is more like a heuristic adjustment for the calculation scale of the advantages of Vine PPO. (1) It does not provide a good explanation for why the paragraphs formed by fixed-length tokens represent different semantics. (2) It does not explain why dividing paragraphs according to token probabilities is feasible and related to semantics.
Overall, the author has provided an intuitive and effective form of advantage calculation and gradient calculation. If the author can address all the concerns raised by the reviewers with high quality, I will consider raising the score.

---

> ### Author Rebuttal · Authors · 2025-07-31
>
> Thanks for your valuable and constructive comments. Below, we provide detailed responses to the concerns and questions you have raised.
>
> > W1: The design details of the method are rather detailed, but the design motivation is too vague and empirical. It is more like a heuristic adjustment for the calculation scale of the advantages of Vine PPO..
>
>
> Thanks for this feedback. The motivation of SPO is to find a new way to solve the critical credit assignment problem in RL for LLM, which is not well solved by the existing token-level and trajectory-level approaches. Our main insight is to adopt segment-level (medium-level) advantage estimation. This approach provides richer feedback signals than GRPO and eliminates the need for a critic model. However, the segment-level approach introduces new problems: (1) how to perform the segment partition? (2) how to compute the segment-level advantage efficiently, especially for long CoT? (3) how to do policy update within each segment? The solutions proposed in VinePPO are inadequate to solve these problems. Compared to VinePPO, our SPO is more general, effective, and efficient, as described below:
>
> 1. **Generality:** VinePPO uses semantic partitioning, i.e., splitting the response into steps based on semantic cues such as line breaks. This approach is limited to scenarios where the model’s responses can be easily divided into semantically meaningful steps. However, for many tasks, such as code generation or complex reasoning, it is challenging or even impossible to identify such semantic boundaries, which severely limits the applicability of VinePPO. In contrast, our method supports flexible segment partitioning without requiring semantic completeness, making it applicable to almost any task. Moreover, our framework allows seamless adjustment of segment granularity, from token-level (i.e., PPO) to trajectory-level (i.e., GRPO). Essentially, VinePPO can be regarded as a special case of our SPO-chain, using a straightforward semantic partitioning strategy without probability guidance.
>
> 2. **Effectiveness:** Compared to VinePPO, our method introduces a novel probability-guided segment partition strategy and a probability-mask technique. These allow us to partition the response at positions where the $V$ value is more likely to change, and to focus the optimization on critical tokens. Experimental results show that our method not only reduces sampling cost but also achieves higher accuracy than VinePPO. Furthermore, we analyze from a variance perspective and explain why using Monte Carlo estimation for segment advantage computation is effective.
>
> 3. **Efficiency:** Both VinePPO and our SPO-chain incur substantial sampling costs when dealing with long CoT, making them impractical for such scenarios. To address this, we propose SPO-tree, a novel tree-based structure that significantly improves efficiency and enables our method to be effectively applied in long CoT settings.
>
> > W2: It does not provide a good explanation for why the paragraphs formed by fixed-length tokens represent different semantics. (2) It does not explain why dividing paragraphs according to token probabilities is feasible and related to semantics.
>
> Thanks for your comments. In our method, segment advantage values are estimated using Monte Carlo sampling rather than being predicted by a reward model. As a result, our SPO does not require each segment to have coherent or distinct semantics, nor do we rely on explicit semantic boundaries. Instead, the main objective of our segmentation is to ensure that the value function ($V$-value) at segment boundaries is likely to change, so that the computed advantage is non-zero. A non-zero advantage indicates whether generating the segment increases or decreases the probability of task success, thereby providing meaningful learning signals for model optimization. In the short chain-of-thought scenario, we identify tokens with low generation probability as cutpoints, which often correspond to positions where the model’s reasoning trajectory may branch and thus where the $V$-value may change. We then segment the response at regular intervals of these cutpoints, increasing the likelihood that the $V$-values at segment boundaries are different. In the long chain-of-thought scenario, each segment naturally contains a large number of cutpoints. Therefore, using a fixed-token-count partition strategy is a reasonable simplification for this setting, which also makes it compatible with our tree-based sampling method.
>
> > Q1: As VinePPO is the main comparison method, did it also set the same number of MC rollout to ensure the fairness of the experiment?
>
> Yes, we use the same number of rollouts (K=9) for both VinePPO and our SPO-chain. We will clarify it in the experimental setup.
>
> > Q2: The basic model is too simple and has already been distilled using R1's mathematical data in advance. This makes the base model itself already have very good scores in mathematics (as shown in the author's experimental comparison table). The author should conduct experiments and verification in 1.5-base or qwen-1.6b-base, align with other RL-zero papers.
>
> Thank you for your valuable suggestion regarding the evaluation on base models. We address your concern in two parts:
>
> **Motivation for using distilled models:** Our primary motivation for choosing R1-distilled models is to evaluate our method’s effectiveness in long CoT scenarios. Specifically, the R1-Distill model demonstrates significantly longer output sequences compared to base or instruct models, which enables assessment of our approach in terms of both efficiency and model accuracy in long CoT reasoning settings. Furthermore, there are many recent works that perform RL training on R1-distilled models (e.g. DeepScaleR and STILL-3 in Table 1).
>
> **Experiments on Base Models:** According to your suggestion, we have additionally conducted experiments on the MATH dataset using the Qwen2.5-Math-1.5B (base) model, which is also used in the recent zero-RL papers like Dr. GRPO. The results are listed below:
>
> Qwen2.5-Math-1.5B (base):
> | | | |
> |:-:|:-:|:-:|
> |Training Steps / Accuracy (%)|GRPO|SPO-tree|
> |100|**68.4**|67.6|
> |200|70.4|**71.4**|
> |300|71.4|**71.8**|
> |400|71.4|**72.8**|
> |500|71.6|**73.6**|
> |600|72.4|**74.2**|
> |700|73.0|**74.2**|
> |800|73.0|**74.2**|
>
> In addition, we evaluated our approach with both Qwen2.5-0.5B-Instruct and Qwen2.5-1.5B-Instruct models on the GSM8K dataset. The results are shown below:
>
> Qwen2.5-0.5B-Instruct:
> | | | |
> |:-:|:-:|:-:|
> |Training Steps / Accuracy (%)|GRPO|SPO-tree|
> |100|56.3|**57.8**|
> |200|57.5|**61.1**|
> |300|57.5|**62.0**|
> |400|58.0|**62.4**|
> |500|59.9|**62.9**|
> |600|59.9|**64.1**|
>
> Qwen2.5-1.5B-Instruct:
> | | | |
> |:-:|:-:|:-:|
> |Training Steps / Accuracy (%)|GRPO|SPO-tree|
> |100|74.6|**79.1**|
> |200|77.5|**81.7**|
> |300|77.5|**82.6**|
> |400|78.1|**83.5**|
> |500|78.1|**83.5**|
> |600|78.1|**84.0**|
>
> These results demonstrate that SPO-tree consistently outperforms GRPO across base, instruct, and R1-distill models, as well as different datasets.
>
>
> > Q3: It seems that the actual response generation of LLM has not been released.
>
> Thank you for your comment. We have provided examples of model outputs in the previously submitted Appendix, including a short CoT exmaple in Appendix I with cutpoints and segment demonstration, as well as a long CoT output in Appendix J.

---

> > ### Author Response · Authors · 2025-08-08
> >
> > Thank you very much for taking the time to review our submission and for your insightful comments. We have carefully responded to all the concerns you raised in your previous review and have provided additional experimental results. As the discussion phase is coming to an end, we want to kindly confirm whether our responses have resolved your concerns, or if there are any points you would like us to further clarify. If you feel that our responses have satisfactorily addressed your concerns, we would sincerely appreciate your consideration for a higher score. Thank you again for your time and support.

---

### Official Review · Reviewer_3Avs · 2025-07-02

**Clarity:** 3
**Significance:** 2
**Originality:** 3
**Rating:** 4
**Confidence:** 4

**Summary:**

This paper proposes Segment Policy Optimization (SPO) for training LLM reasoners with better credit assignment. It cuts the CoT into several segments based on token probability values, then conducts monte carlo estimation of the value of each segment by unrolling the current policy in either chain or tree format. Finally, the policy gradient is applied with a probability mask to only train those "critical" tokens. Empirical results show that the proposed method achieves better learning efficiency and final performance compared to baselines including VinePPO and GRPO.

**Questions:**

(1) line 297 - 299 & fig4: int2 is better than int5 from the figure, but the text says "whereas an overly coarse interval can severely degrade accuracy", which is confusing to me

(2) following (1), i am curious what the result of int1 will be?

(3) how did you ensure that the comparison with GRPO is fair? for example, given a question, GRPO unrolls 8 responses, while the proposed method unrolls 8 responses + K samples for MC value estimation. GRPO could do similar thing: given the 8 responses, keep the thinking process and resample the final answer K times to estimate the value (this probably can be int1 mentioned in (2)). have you considered this setting for a fairer comparison?

(4) have you tried SPO-chain on the long CoT scenario? or SPO-tree on the short CoT scenario?

(4) for fig5a, have you tried GRPO with probability mask? i am very curious its wall-clock time curve

**Ethical Concerns:**

["NO or VERY MINOR ethics concerns only"]

**Final Justification:**

This paper presents a bag of techniques to improve the original VinePPO method.

I would suggest a weak accept given the reasonable motivation of the proposed techniques and good empirical results.

**Limitations:**

yes

**Quality:**

2

**Strengths And Weaknesses:**

strengths:
- the paper is clearly written and all techniques are simple to understand
- efficiency-wise the proposed method is better than vinePPO



weakness:
- line 357: i do not think evaluate DeepScaleR under 2k or 4k budget is a fair comparison, since it is trained with much longer context size. it is also not directly related to token efficiency because of that reason. but GRPO indeed leads to redundancy and token inefficiency, as suggested by [1].
- the experiments did not show vineppo with probability mask, which i think is an important baseline to claim that the proposed method does better credit assignment



[1] https://arxiv.org/pdf/2503.20783

---

> ### Author Rebuttal · Authors · 2025-07-31
>
> Thanks for your valuable and constructive comments. Below, we provide detailed responses to the concerns and questions you have raised.
>
> > W1: line 357: I do not think evaluating DeepScaleR under 2k or 4k budget is a fair comparison, since it is trained with much longer context size. it is also not directly related to token efficiency because of that reason. But GRPO indeed leads to redundancy and token inefficiency, as suggested by [1].
>
> Thank you for pointing out this issue. We agree that evaluating DeepScaleR under a 2K or 4K context budget is not a fair comparison, as it was trained with a much longer context length and on a larger dataset. We plan to remove the DeepSacleR result from Table 1. To make our comparisons fair and meaningful, we focus primarily on GRPO and SPO models that are trained with the same context length and the same dataset. As shown in Table 1, across 2K, 4K, and 32K context settings, models trained with SPO consistently achieve higher accuracy than those trained with GRPO. This demonstrates the effectiveness of the SPO.
>
> > W2: The experiments did not show VinePPO with a probability mask, which I think is an important baseline to claim that the proposed method does better credit assignment
>
> Thank you for your suggestion to include the results of VinePPO with the probability mask. In our new experiments, after adding our probability mask strategy, VinePPO improves its accuracy from 53.4% to 55.6%, demonstrating that our probability-mask strategy is generally effective. However, our SPO-chain (int5) method still achieves a higher accuracy of 56.7%, which indicates that our probability-guided segment partition strategy provides better credit assignment than VinePPO’s semantic split. Furthermore, as illustrated in Figure 3(b), our SPO-chain uses fewer segments and results in shorter training time compared to VinePPO, also highlighting its better efficiency than VinePPO.
>
> > Q1: line 297 - 299 & fig4: int2 is better than int5 from the figure, but the text says "whereas an overly coarse interval can severely degrade accuracy", which is confusing to me
>
> Thank you for your comments and for pointing out this potential confusion regarding the interval settings. We would like to clarify the meaning of "int2", "int5", and "int100" in our experiments as follows:
>
> - "int2" means partitioning the model output at intervals of every 2 cutpoints, which results in the most (finest) segments.
> - "int5" means partitioning the model output at intervals of every 5 cutpoints, resulting in fewer (coarser) segments compared to int2.
> - "int100" is an extreme setting: We partition the model output at intervals of every 100 cutpoints, which, given the total number of cutpoints, results in only one segment.
>
> As Figure 4(a) shows, both int2 and int5 achieve high and similar accuracy, with int2 offering a marginal improvement. This suggests that an excessively fine-grained interval (e.g., int2 versus int5) provides limited benefits. However, int100 (only one segment) exhibits a significant performance drop, which supports our statement that “an overly coarse interval can severely degrade accuracy.”
>
> > Q2: following (1), I am curious what the result of int1 will be?
>
> We suppose that int1 refers to only a single segment in your question. If so, this setting corresponds to what we refer to as int100 in our experiments (given the total number of cutpoints, int100 results in only one segment).  As shown in Figure 4(a), the performance under this setting is significantly degraded, demonstrating the negative impact of having an overly coarse interval, as discussed above.
>
> > Q3: How did you ensure that the comparison with GRPO is fair? For example, given a question, GRPO unrolls 8 responses, while the proposed method unrolls 8 responses + K samples for MC value estimation. GRPO could do a similar thing: given the 8 responses, keep the thinking process and resample the final answer K times to estimate the value (this probably can be int1 mentioned in (2)). Have you considered this setting for a fairer comparison?
>
> Thank you for this insightful and thoughtful suggestion. We would like to clarify that the method you described, while interesting, does not align with the standard GRPO setting, as commonly used in previous works and open-source frameworks.
>
> As mentioned by the reviewer, the reviewer's proposed method can actually be seen as a special case of our SPO-tree (specifically "8-K") with two layers: the first layer contains 8 nodes, and each node has K leaves. The main difference lies in how the response is partitioned: the reviewer's proposed method partitions the response  at \</think\> tokens, whereas our SPO partitions the response by fixed token count rather than specific delimiters like \</think\>.
>
> We compare the test accuracy of the reviewer's method and SPO-tree (6-6-6) as follows.  As shown in the table below, the accuracy of the reviewer’s proposed method is lower than that of our SPO-tree (6-6-6). This may be because using only two segments provides less feedback than three segments, and strictly segmenting at \</think\> may not be optimal, since the final answer's quality is heavily dependent on the preceding thinking process.
>
> | | | |
> |:-:|:-:|:-:|
> |Training hours / Accuracy (%)|2 segments split at \</think\>|SPO-tree (6-6-6)|
> |2|63.8|**66.0**|
> |4|69.6|**72.4**|
> |6|71.0|**75.0**|
> |8|74.2|**76.0**|
> |10|74.2|**76.2**|
>
> > Q4: Have you tried SPO-chain on the long CoT scenario? or SPO-tree on the short CoT scenario?
>
> **SPO-tree for short CoT:** Yes, we have conducted experiments with SPO-tree in the short CoT scenario, as shown in Figure 3(a). The results show that SPO-tree achieves 56.4% test accuracy, which is comparable to SPO-chain (56.7%) and outperforms all other baselines. This validates the effectiveness of SPO-tree  in short CoT scenarios and shows that it can serve as an efficient alternative.
>
> **SPO-chain for long CoT:** For the long CoT scenario, we did not evaluate SPO-chain, due to its computational inefficiency in this scenario. Specifically, at each segment boundary, SPO-chain samples K independent trajectories for value estimation. As each trajectory can be quite long in the long CoT scenario (i.e., involving thousands of tokens), the overhead for value estimation becomes prohibitively large, making it impractical for long CoT settings. Moreover, in SPO-chain, the samples used for value estimation are simply discarded, leading to substantial sample waste in the long CoT scenario.
>
> The above issue inspired us to develop SPO-tree for long CoT. By organizing the sampling process into a tree structure and aggregating rewards from the bottom up, SPO-tree enables all samples used for value estimation can also contribute to policy optimization, substantially reducing the wastage inherent in SPO-chain. This design choice enables SPO-tree to be effectively applied to long CoT scenarios.
>
> > Q5: For fig5a, have you tried GRPO with probability mask? i am very curious its wall-clock time curve.
>
> Thank you for your interest in this question. As suggested, we conduct new experiments by adding probability mask to GRPO, using the same configuration as GRPO in Fig. 6(a). The results are shown in the table below.
>
> We observe that GRPO with probability mask outperforms vanilla GRPO to some extent, which demonstrates the effectiveness of the probability mask in long CoT scenarios. However, SPO-tree still achieves a significant advantage compared to GRPO with probability mask. This indicates that introducing the segment advantage value is crucial for achieving better performance.
>
> | | | | |
> |:-:|:-:|:-:|:-:|
> |Training hours / Accuracy (%)|Vanila GRPO|GRPO with probability mask|SPO-tree|
> |10|62.6|63.0|**63.2**|
> |20|65.0|65.4|**71.2**|
> |30|65.0|68.2|**71.2**|
> |40|65.6|68.2|**75.0**|
> |50|66.0|68.2|**75.0**|
> |60|66.2|68.2|**75.8**|

---

> > ### Comment · Reviewer_3Avs · 2025-08-05
> >
> > Thank the authors for the detailed rebuttal and newly added experiment results.
> >
> > After checking the results I have a few thoughts:
> >
> > 1) The overall method (segment and get fine-grained advantage estimation to improve credit assignment) is very similar to VinePPO without the probability mask (please correct me if I misunderstood).
> > 2) The probability mask empirically improves GRPO a lot (Fig4c).
> > 3) Combining 1) and 2), I would expect VinePPO with prob mask to be mechanistically similar to your method and empirically similar as well. Could you comment on your results (VinePPO 55.6% v.s. Yours 56.7)? What brings advantage to the proposed method? Or could it be randomness during evaluation or some configuration mismatch during training? I would love to understand it more rigorously.
> >
> > Thanks for your time in advance!

---

> > > ### Author Response · Authors · 2025-08-06
> > > **Title: (1/2) Response to Reviewer 3Avs's Additional Questions**
> > >
> > > Thanks for your thoughtful questions and for providing us the opportunity to further clarify our work.
> > >
> > > **We provide our detailed response in two parts.**
> > >
> > > >1. The overall method (segment and get fine-grained advantage estimation to improve credit assignment) is very similar to VinePPO without the probability mask (please correct me if I misunderstood).
> > >
> > > There are significant differences between our method and VinePPO both in terms of core ideas and technical implementations. These differences are detailed as follows:
> > >
> > > **Idea differences:** The key idea of our method is to introduce flexible segment-level (medium-level) advantage to overcome the limitations of both coarse-grained methods like GRPO and fine-grained methods like PPO, as well as to unify them within a single framework. In contrast, VinePPO paper does not discuss the limitations of GRPO and PPO from the perspective of advantage estimation granularity, nor does it recognize that dividing the model's response into multiple segments and introducing segment-level advantages is the key approach to addressing these problems. In fact, VinePPO does not introduce the concept of "segments"—it simply assumes that the response has already been semantically partitioned into steps, and focuses on estimating the advantage at each step using MC methods. It is important to note that **our definition of segments is much more general than the “semantic steps” used in VinePPO**, so our method is not simply “VinePPO without the probability mask.”
> > >
> > > **Technical differences:** Compared to VinePPO, we introduce the concept of segments and employ different/novel partitioning and advantage estimation strategies, **making our method more general, effective, and efficient.**
> > >
> > > 1. **Segment Definition and Partitioning:** VinePPO estimates the advantage at semantic steps, which are determined by semantic cues such as line breaks. This approach is restrictive because it only works in scenarios where responses can be easily divided into meaningful semantic steps. For many tasks (e.g., code generation, complex reasoning), it is difficult or infeasible to define such boundaries, which greatly limits VinePPO’s applicability.  In contrast, **our method defines a segment as any consecutive sequence of tokens, without requiring semantic completeness**. This enables (1) **much broader applicability**, allowing our method to handle tasks without clear semantic steps; (2) **seamless adjustment of segment granularity**, from token-level (PPO) to trajectory-level (GRPO); and (3) **the design of more effective segmentation strategies**, such as the cutpoint-based approach used in SPO-chain.
> > >
> > > 2. **Advantage Estimation:** VinePPO's simple advantage estimation strategy incurs high sampling costs, making it infeasible for long CoT scenarios. To address this, **we propose SPO-tree, a novel tree-structured sampling and advantage estimation approach that greatly improves efficiency and makes MC-based advantage estimation practical. This innovation enables our method to be effectively applied to long CoT scenarios.**
> > >
> > > We have provided extensive experimental evidence to support these points:
> > > 1. Our segmentation method (e.g., SPO-chain with cutpoint-based segmentation) achieves better accuracy than VinePPO without the probability mask (see Figure 4(c)), and also outperforms VinePPO when both methods use the probability mask.
> > >
> > > 1. VinePPO is impractical for long CoT tasks, whereas our SPO-tree allows efficient training and shows superior test accuracy compared to GRPO within the same training time, as shown in Figure 5(a).
> > >
> > > **In fact, VinePPO can be viewed as a special case within our framework, corresponding to step-based segmentation, chain-like advantage estimation and policy optimization without probability mask.** Within our framework, we have developed more advanced methods (e.g., SPO-chain, SPO-tree) that offer broader applicability, better effectiveness and efficiency.
> > >
> > > >2. The probability mask empirically improves GRPO a lot (Fig4c).
> > >
> > > This statement is correct in short chain-of-thought scenarios. Adding the probability mask leads to substantial improvement for GRPO, as shown in Figure 4(c). However, in long CoT settings, as demonstrated in our rebuttal experiments, although the probability mask does provide a performance boost to GRPO, there is still a significant gap compared to SPO-tree. This result suggests that introducing segment-level advantage values is crucial for achieving superior performance, especially when handling longer and more complex outputs.
> > >
> > > **We address your question 3 in part 2. Please refer to the comments below.**

---

> > > > ### Author Response · Authors · 2025-08-06
> > > > **Title: (2/2) Response to Reviewer 3Avs's Additional Questions**
> > > >
> > > > >3. Combining 1) and 2), I would expect VinePPO with prob mask to be mechanistically similar to your method and empirically similar as well. Could you comment on your results (VinePPO 55.6% v.s. Yours 56.7)? What brings advantage to the proposed method? Or could it be randomness during evaluation or some configuration mismatch during training? I would love to understand it more rigorously.
> > > >
> > > > Thank you for your thoughtful question. By combining 1) and 2), VinePPO with the probability mask still has lower accuracy than our SPO. A short answer for the reason is that our cutpoint-based segment partition strategy is more effective and provides better credit assignment than VinePPO's semantic partition. This improvement is not caused by randomness or other configuration issues. A detailed reason is as follows:
> > > >
> > > > (1) **SPO's better credit assignment**: Step-based segmentation, as used in VinePPO, can lead to segments where the transition probabilities between tokens are all close to 1 within a step.  In these cases, the value function at the segment boundaries barely changes, resulting in advantage values close to zero, which provides no learning signal for the model. In SPO-chain, we adopt a cutpoint-based segmentation strategy. This strategy significantly increases the likelihood that the value ($V$) at the start and end of each segment will be different, resulting in non-zero advantages. Non-zero advantages indicate that generating the tokens in this segment increases or decreases the probability of final success, thus providing effective credit assignment and meaningful learning signals to the model.
> > > >
> > > > (2) **Not caused by randomness or other configuration issues**: We use the same number of rollouts (K=9) for both VinePPO and our SPO-chain. Our experimental results in different settings consistently confirm our advantage: Our method achieves better accuracy than VinePPO even without the probability mask (see Figure 4(c)), and still outperforms VinePPO when both methods use the probability mask. Furthermore, our method achieves better performance while using fewer segments than VinePPO, as shown in Figure 3(b). This improvement is due to our more general segment definition and more effective partitioning strategy, rather than randomness or any configuration mismatch.
> > > >
> > > > Thank you again for your valuable questions and comments. We are happy to discuss any additional concerns you might have.

---

> > > > > ### Comment · Reviewer_3Avs · 2025-08-06
> > > > >
> > > > > Thank you for providing detailed responses.
> > > > >
> > > > > I think from token-level MDP perspective, VinePPO is a general idea that we can branch out from any (token) step and do MC rollouts for value estimation (see the old TRPO paper [1]). This is for getting a better credit assignment than a learned critic, which is the same objective as this work.
> > > > >
> > > > > The VinePPO paper used semantic step as the cutpoint while you used fixed token count or probability accumulation, with one is worse than VinePPO and the other is better. I personally think this is not fundamentally different and lie within the scope of VinePPO (they are implementation details but not algorithmic innovations).
> > > > >
> > > > > Nevertheless, I think the intuition makes sense: if the likelihood of the whole sequence is near 1, then the value estimation at both ends are likely to be the same thus no advantage in sparse reward setting. This should be the source of the improvement if we disable the probability mask technique?
> > > > >
> > > > > Overall, I think this paper has many small technical changes on top of VinePPO which make it work better empirically. Given the good experimental results and the detailed clarifications, I will raise my score to positive.
> > > > >
> > > > > Thanks again!
> > > > >
> > > > > ---
> > > > >
> > > > > [1] https://arxiv.org/pdf/1502.05477

---

> > > > > > ### Author Response · Authors · 2025-08-07
> > > > > >
> > > > > > Thank you very much for your thoughtful comments and for taking the time to review our responses. We sincerely appreciate your constructive feedback and your decision to increase the score.
> > > > > >
> > > > > > Your understanding of the source of improvement brought by our approach when the probability mask technique is disabled is correct.  As you noted, in VinePPO’s step-based branching, there can be cases where the likelihood of the whole sequence is near 1. In these cases, the estimated values at both ends will be the same, resulting in zero advantages. Our cutpoint-based segmentation, on the other hand, significantly increases the probability of value changes at segment boundaries, thus generating more segments with non-zero advantages. This provides more informative training signals and is indeed a key source of the observed improvements.
> > > > > >
> > > > > > We would also like to add a further clarification about the generality of our segment-based framework compared to VinePPO’s step-based approach. In VinePPO, the advantage is estimated at each step and then attributed to all tokens within that step.  In contrast, our method introduces the general concept of "segment", defined as any span of consecutive tokens. **This enables segmentation at arbitrary positions and allows the use of any number of segments within a trajectory, providing the flexibility to adjust the granularity as needed.** Such flexibility brings two significant benefits:
> > > > > >
> > > > > > **More effective segmentation:** As discussed above, this general framework enables us to design more effective segmentation strategies, such as the cutpoint-based approach. This approach can generate more segments with non-zero advantages, thereby achieving better credit assignment. Our experiments demonstrate that our method delivers improved performance compared to VinePPO while also reducing sampling cost, highlighting both the effectiveness and efficiency of our segmentation approach.
> > > > > >
> > > > > > **More efficient advantage estimation:** VinePPO’s approach relies on costly chain-like MC rollouts, which is a critical limitation and makes it impractical for long sequences. **In contrast, the flexibility of our framework—allowing partitioning at arbitrary positions and supporting any number of segments—enables us to design a novel tree-structured sampling strategy. This greatly improves estimation efficiency and allows our method to be effectively applied even in long CoT scenarios.**
> > > > > >
> > > > > > In summary, **our generalized segment-based approach—allowing partitioning at arbitrary positions and using any number of segments—provides a foundation for more effective and efficient techniques, as demonstrated by our empirical results**.
> > > > > >
> > > > > > Thank you again for your careful reading, recognition of our work, and for giving us the opportunity to clarify these points. Your feedback has been extremely valuable for us,  and we sincerely appreciate your positive evaluation.

---

### Official Review · Reviewer_wXY9 · 2025-07-06

**Clarity:** 3
**Significance:** 3
**Originality:** 3
**Rating:** 5
**Confidence:** 4

**Summary:**

The paper introduces Segment Policy Optimization—an algorithm aiming to have fine-grained advantage estimation similar to PPO without introducing challenges in value estimation for a token-level critic. Therefore, SPO is partially along the spectrum towards GRPO, but instead of providing reward signal only from the termination of a trajectory, SPO partitions a trajectory into several segments, and computes advantage at the last “splitting token” of each segment. The paper offers an intelligent segment splitting strategy at low-probability tokens, and performs policy optimization over variable-length segments. Empirical results indicate boosts over GRPO in both short chain-of-thought and longer reasoning settings.

**Questions:**

Note on intro: PPO can be implemented using shared backbone for actor and critic, so the  requirement for a separate critic model in Line 40 isn’t accurate.

All experiments are performed on mathematical benchmarks—so it would be great to see experiments in non-math settings to see the generality of the methods.

It would be great to have a longer discussion of limitations of the method—what are some settings where segment-splitting might not be as effective as having token-level advantages from PPO? Or where is value estimation too challenging and GRPO a superior option?

**Ethical Concerns:**

["NO or VERY MINOR ethics concerns only"]

**Final Justification:**

I initially had concerns that results were limited to math settings, and was unclear on how to motivate the use of SPO in novel settings, as opposed to GRPO or PPO. In the rebuttal, the authors demonstrate additional experimental results beyond mathematical settings, and additionally provide reasoning on scenarios to use the different RL algorithms. As a result, I raised my score.

**Limitations:**

Yes

**Quality:**

3

**Strengths And Weaknesses:**

**Strengths**

The paper crafts an insightful problem setup in juxtaposing the relative strengths and weaknesses of PPO and GRPO. The probability-guided segment splitting strategy is novel, and in particular focuses on the most “important” decisions in a trajectory that can be informative.

Empirical results convincingly show the efficacy of both SPO-Chain and SPO-Tree. Ablations are quite valuable—as it is interesting to see that SPO-Tree performance isn’t very affected by hyperparameters tested in Figure 5.

Ablations are also performed on the key segment-length parameter.

The implementation of both chain and tree-based segment advantage estimation is a nice intuitive tradeoff.

**Weaknesses**

Note on intro: PPO can be implemented using shared backbone for actor and critic, so the  requirement for a separate critic model in Line 40 isn’t accurate.

All experiments are performed on mathematical benchmarks—so it would be great to see experiments in non-math settings to see the generality of the methods.

It would be great to have a longer discussion of limitations of the method—what are some settings where segment-splitting might not be as effective as having token-level advantages from PPO? Or where is value estimation too challenging and GRPO a superior option?

---

> ### Author Rebuttal · Authors · 2025-07-31
>
> Thanks for your valuable and constructive comments. Below, we provide detailed responses to the concerns and questions you have raised.
>
> >  Q1: Note on intro: PPO can be implemented using shared backbone for actor and critic, so the requirement for a separate critic model in Line 40 isn’t accurate.
>
> Thanks for pointing out this issue. We will change the statement in Line 40 from "Additionally, PPO requires maintaining a separate critic model, typically as large as the actor, which doubles the memory footprint and computational cost, making it less efficient for large-scale training" to “Additionally, PPO employs either a separate critic model or an additional critic head to predict value function, leading to extra memory and computation overhead”.
>
> > Q2: All experiments are performed on mathematical benchmarks—so it would be great to see experiments in non-math settings to see the generality of the methods.
>
> Thank you for your insightful suggestion regarding the evaluation of our methods on non-mathematical benchmarks to assess their generality. Due to limited time and computational resources, we have not yet conducted experiments on other computationally heavy non-math tasks. We recognize the importance of verifying the general applicability of our approach and plan to include evaluations on non-mathematical tasks in our future work. We appreciate your suggestion and believe that extending our evaluation beyond mathematical tasks will further demonstrate the versatility and robustness of our method.
>
> > Q3: It would be great to have a longer discussion of limitations of the method—what are some settings where segment-splitting might not be as effective as having token-level advantages from PPO? Or where is value estimation too challenging and GRPO a superior option?
>
> Thanks for the suggestions. We will discuss more limitations of our method in the future version as follows.
>
> Our SPO framework provides a unified perspective that encompasses both GRPO (when segment length equals the entire response) and PPO (when segment length is set to 1). This design allows SPO to flexibly adapt to the characteristics of various tasks by tuning the segment granularity and advantage estimation method.
>
> **PPO better than SPO:** PPO is particularly effective in environments where the agent frequently revisits similar or identical states during training. A classic example is traditional reinforcement learning tasks, such as playing a video game, where the environment resets and comparable states are encountered repeatedly. This frequent revisitation enables the critic model to learn and refine accurate predictions for the same or similar states, allowing for fine-grained, token-level feedback to guide learning. In such scenarios, fine-grained token-level supervision, as provided by PPO, is more suitable.
>
> However, in scenarios like LLM-based mathematical problem solving, different prompts often lead to entirely different trajectory distributions. This means that the agent rarely sees similar states across examples. Consequently, the critic model cannot reliably predict value estimates for states it seldom encounters, which reduces PPO's effectiveness. In these cases, critic-free approaches like GRPO or SPO may be more effective.
>
> **GRPO better than SPO:** For very short responses, further segmenting can result in segments containing only a few tokens, where the value at the start and end of each segment may be essentially the same. This can lead to unnecessary splitting and introduce extra computational overhead for advantage estimation. In such cases, the simplicity of GRPO, which relies solely on the overall reward, leads to better computational efficiency.

---

> > ### Author Response · Authors · 2025-08-02
> > **More information about experiments on the non-math task**
> >
> > We would like to thank the reviewer again for the valuable suggestion to evaluate our method on non-mathematical settings in order to demonstrate its generality. As we mentioned in our rebuttal, due to limited time and computational resources, we were unable to include experiments on non-mathematical tasks before the rebuttal deadline.
> >
> > We understand that this response comes after the designated rebuttal period, but for the sake of completeness—and in response to your valuable suggestion—we report here our new experimental results on the Knights-and-Knaves dataset. This is a classic logic puzzle benchmark where each character is either a knight (who always tells the truth) or a knave (who always lies), and the goal is to deduce each character’s identity based on their statements. Specifically, we used the 3-people (3ppl) subset and trained our models with Qwen-2.5-1.5B-Instruct. The results are shown in the table below:
> >
> > | Step | GRPO      | SPO-tree (6-6-6) |
> > | ---- | --------- | ---------------- |
> > | 20   | 0.060     | **0.230**        |
> > | 40   | **0.585** | 0.580            |
> > | 60   | **0.655** | 0.580            |
> > | 80   | 0.690     | **0.840**        |
> > | 100  | 0.725     | **0.865**        |
> > | 120  | 0.725     | **1.110**        |
> > | 140  | 0.725     | **1.180**        |
> > | 160  | 0.725     | **1.180**        |
> > | 180  | 0.725     | **1.180**        |
> > | 200  | 0.760     | **1.215**        |
> > | 220  | 0.795     | **1.215**        |
> > | 240  | 0.795     | **1.250**        |
> > | 260  | 0.860     | **1.250**        |
> > | 280  | 0.935     | **1.285**        |
> > | 300  | 0.935     | **1.355**        |
> >
> > At step 300, our method (SPO-tree) achieved a test reward of 1.355, outperforming the GRPO baseline, which reached 0.935. These results confirm that our approach not only performs well on mathematical tasks, but also generalizes effectively to non-mathematical logical reasoning problems. We sincerely appreciate your suggestion, which has helped us to further improve and strengthen our work.

---

> > > ### Comment · Reviewer_wXY9 · 2025-08-04
> > >
> > > Thanks for the thorough rebuttal and responses.
> > >
> > > The elaboration on situations where GRPO and PPO are respectively appropriate is quite helpful--and helps frame that there is in fact repeated visitation to similar scenarios at a segment-level, but not at a token level, for LLM tasks.
> > >
> > > The new experimental results on Knights and Knaves are definitely worth including in the final version of the paper--I think it's particularly for the community to see how results in RLVR generalize beyond math tasks. Thank you for running this experiment.
> > >
> > > After reading this rebuttal, I have raised my score.

---

### Note · Authors · 2025-08-12

We sincerely thank the AC and reviewers for their valuable feedback. We addressed all concerns and added new experiments.

- **Beyond math.** Our approach performs well on non-math datasets.
- **Different from VinePPO**. VinePPO's method of estimating advantages via costly **per-step, K-fold full-trajectory resampling** has critical limitations: dependency on pre-defined semantic steps limits its generality, and heavy sampling overhead renders it inefficient and thus impractical for long CoT scenarios. We start from the granularity of advantage estimation and **propose a generalized framework for segment-level advantage estimation**, unifying token- and trajectory-level methods via a continuous granularity spectrum and supporting arbitrary partition points and segment counts. On this foundation, we develop more advanced techniques, such as **cutpoint-based partitioning**, which does not rely on semantic steps and yields more segments with non-zero advantages, improving credit assignment; and **tree-based sampling**, where **segment advantages are estimated *concurrently* with trajectory generation**. This obviates the need for VinePPO’s costly resampling to estimate advantage, thereby substantially improving efficiency and making our approach effective in long CoT scenarios.
- **Broader validation.** On base, instruct, and R1-distill models across multiple datasets, SPO-tree consistently outperforms GRPO.
- **Granularity choice.** Under equal wall-clock time, moderate intervals offer the best accuracy–efficiency trade-off, supporting the effectiveness of **segment-level (medium-level)** advantages.

**Regarding the concerns from Reviewer vcBi**, we have provided a detailed rebuttal to address her/his concerns. Although the reviewer had previously indicated she/he would raise the score if these concerns were resolved, we have not received further feedback during the discussion phase. We therefore respectfully ask the AC to consider our response and added results when interpreting this score.

In summary, our SPO framework unifies token-level (PPO) and trajectory-level (GRPO) approaches, addressing critic unreliability and overly coarse credit assignment. Built upon this foundation, we introduce several advanced techniques, including **cutpoint-based partitioning, tree-based sampling, and a novel probability mask technique**. Our approach delivers consistent improvements across diverse models and datasets, enabling more effective RL for LLMs.

---

### Decision · Program_Chairs · 2025-09-17

**Decision:**

Accept (poster)

**Comment:**

The paper introduces Segment Policy Optimization—an algorithm aiming to have fine-grained advantage estimation similar to PPO without introducing challenges in value estimation for a token-level critic. Therefore, SPO is partially along the spectrum towards GRPO, but instead of providing reward signal only from the termination of a trajectory, SPO partitions a trajectory into several segments, and computes advantage at the last “splitting token” of each segment. The paper offers an intelligent segment splitting strategy at low-probability tokens, and performs policy optimization over variable-length segments. Empirical results indicate boosts over GRPO in both short chain-of-thought and longer reasoning settings.

Reviewers appreciate the motivation and technical insights of the proposed method. The empirical evaluation is also comprehensive (after adding more during the rebuttal). The AC believes this paper makes a valuable contribution, and thus recommends acceptance.